# The scaffold protein Nde1 safeguards the brain genome during S phase of early neural progenitor differentiation

**Shauna L Houlihan[1,2,3], Yuanyi Feng[1,2]\***

[1]Department of Neurology, Northwestern University Feinberg School of Medicine, Chicago, United States; [2]Center for Genetic Medicine, Northwestern University Feinberg School of Medicine, Chicago, United States; [3]Driskill Graduate Program, Northwestern University Feinberg School of Medicine, Chicago, United States

**Abstract** Successfully completing the S phase of each cell cycle ensures genome integrity. Impediment of DNA replication can lead to DNA damage and genomic disorders. In this study, we show a novel function for *NDE1*, whose mutations cause brain developmental disorders, in safeguarding the genome through S phase during early steps of neural progenitor fate restrictive differentiation. Nde1 mutant neural progenitors showed catastrophic DNA double strand breaks concurrent with the DNA replication. This evoked DNA damage responses, led to the activation of p53-dependent apoptosis, and resulted in the reduction of neurons in cortical layer II/III. We discovered a nuclear pool of Nde1, identified the interaction of Nde1 with cohesin and its associated chromatin remodeler, and showed that stalled DNA replication in Nde1 mutants specifically occurred in mid-late S phase at heterochromatin domains. These findings suggest that NDE1-mediated heterochromatin replication is indispensible for neuronal differentiation, and that the loss of NDE1 function may lead to genomic neurological disorders.

**\*For correspondence:** yuanyi-feng@northwestern.edu

**Competing interests:** The authors declare that no competing interests exist.

**Reviewing editor**: Pat Levitt, Children's Hospital Los Angeles, United States

## Introduction

The developmental formation of complex multicellular organs requires the impeccable integration of cell division with differentiation. The precise control of the DNA synthesis (S) phase of each cell division cycle warrants the faithful replication of the entire genome and at the same time establishes the epigenetic state that defines the differentiation identity of individual cells (*McNairn and Gilbert, 2003*; *Nordman and Orr-Weaver, 2012*). Concurrent with genome duplication, the protein components of chromatin are disassembled and re-assembled throughout the S phase into higher order structures that characterize the specific gene expression status of daughter cells (*Alabert and Groth, 2012*; *Budhavarapu et al., 2013*). The task of assuring error-free S phases is especially challenging in the generation of organs with extraordinarily high cell number and diversity, such as the cerebral cortex, during which billions of functionally specialized neurons are produced following a tightly controlled developmental program of cell cycle progression and step-wise neural progenitor fate restriction (*Rakic, 1995*; *Desai and McConnell, 2000*; *Rakic, 2009*; *Florio and Huttner, 2014*). S phase-mediated genome regulation is recently shown to be essential not only for rapid expansion of neural progenitor pool but also for neural differentiation, as the amount of time neural progenitors spend in S phase is highly correlated to the cell fate of cerebral cortical neural progenitor cells (*Arai et al., 2011*). Longer S phase appears more necessary for self renewing than neuron-producing cell cycles, suggesting an S phase specific quality control in maintaining the identity of neural progenitors before their terminal neurogenic division. The importance of genome maintenance in corticogenesis is also underscored by large amounts of clinical and experimental observation, which have shown that the functional

**eLife digest** The brain is a complex organ with many different cell types that each have specialized functions. Mutations in genes that control how the brain develops can have serious consequences—and one of the most important genes involved in the development of the human brain is called *NDE1*.

Individuals who inherit mutated copies of the *NDE1* gene from both parents have brains with fewer folds than a healthy brain, and their brains are up to 90% smaller than normal. These individuals may have serious developmental disabilities, struggle with basic functions like swallowing, and die early. Moreover, having just one defective copy of the *NDE1* gene has been linked to the development of cancers such as leukemia.

The protein encoded by the *NDE1* gene acts as a scaffold for many protein complexes and can be found throughout the cell in various cellular compartments (from the cell periphery to the nucleus). However, it was unclear where the NDE1 protein's activity was most needed in developing brain cells.

Feng and Houlihan now provide a new explanation for *NDE1*'s role in brain development. Looking for molecules that interact with the mouse version of the Nde1 protein revealed that it binds to proteins that control how DNA is packaged inside the nucleus of a cell. In doing so, Nde1 appears to protect the genome of brain stem cells, while these cells' DNA is copied and before they divide to form new cells destined to become neurons.

For a cell to divide, its genetic information must be accurately copied and then segregated between the two new cells to ensure that each receives all the genetic instructions needed to develop and function properly. In brain cells from mice without functional Nde1, the DNA frequently breaks as it is copied. Brain cells that inherit damaged DNA might not function correctly, while serious breaks that affect both strands of the DNA can trigger a response that kills the mutated cell. This means, there are fewer cells that make up the outer layer of the brain, making it less wrinkled than normal. This part of the brain—called the cerebral cortex—is important for many processes including thought and memory, and it helps different areas of the brain communicate with one another. This may explain why mutations in the *NDE1* gene can contribute to a variety of brain disorders.

Errors during DNA replication can also cause cancer to develop. As such, the findings of Feng and Houlihan may also help to explain why some genetic mutations are associated with both cancer and brain disorders.

impairment of genes important for DNA metabolism frequently leads to brain developmental pathology (*McKinnon, 2009*; *Ciccia and Elledge, 2010*; *Zeman and Cimprich, 2014*). However, as the genes involved are also essential for genome surveillance outside of the brain, pathogenic lesions of the brain genome have been believed to associate with the lack of effective DNA damage repair, improper checkpoint signaling, rapid cell proliferation, or increased metabolic, chemical, and physical stresses. It is unclear whether a genome quality control associated specifically with neuronal differentiation is required to ensure the correct genetic and epigenetic identity of both neural progenitors and daughter neurons.

NDE1 is a multifunctional molecular scaffold fundamental for CNS development. It was originally identified as the central nervous system (CNS) specific partner of LIS1 (known as PAFAH1B1) (*Feng et al., 2000*), whose haploinsufficiency results in lissencephaly (smooth brain) (*Reiner et al., 1993*). Homozygous mutations of *NDE1* were found recently to cause microlissencephaly (small and smooth brain) with up to 90% reduction in brain mass, while the affected individuals showed normal development of non-CNS organs (*Alkuraya et al., 2011*; *Bakircioglu et al., 2011*; *Guven et al., 2012*). Moreover, copy number variants (CNVs) in the *NDE1* locus are increasingly shown to associate with a wide spectrum of neuropsychiatric disorders with complex genetic traits (*Ullmann et al., 2007*; *Hannes et al., 2009*; *Heinzen et al., 2010*; *Mefford et al., 2010*; *Nagamani et al., 2011*; *Tropeano et al., 2013*). Genetic epistasis studies in mice demonstrated that Nde1 and Lis1 function synergistically in a dose-dependent manner in governing the generation of late-born cortical neurons that comprise the upper cortical layers II and III. Layer II/III neurons were found specifically reduced in both

Nde1$^{-/-}$ and Nde1$^{+/-}$ Lis1$^{+/-}$ mice, and they were abolished almost completely in Nde1$^{-/-}$Lis1$^{+/-}$ mice along with severe brain hypoplasia but insignificant change in the body size (*Feng and Walsh, 2004*; *Pawlisz et al., 2008*). Neurons in cortical upper layers are evolutionarily novel and undergo great expansion in mammalian evolution (*DeFelipe et al., 2002*; *Molnar et al., 2006*). They are highly diverse projection neurons and essential for cognitive functions including perception, emotion, attention, and memory through making functional connections among various cortical areas and between the two cerebral hemispheres (*DeFelipe et al., 2002*; *Fame et al., 2011*; *Greig et al., 2013*). The essential requirement of Nde1 in the generation of cortical layer II/III neurons underscores the gene dosage dependency of *NDE1* in brain cognition. However, current molecular information on NDE1 (Nde1) is limited by its previously identified association with the cytoskeleton, which does not fully explain its CNS specific phenotype and function.

In the present study of murine models of Nde1 mutations, we report that Nde1 is indispensible for the successful completion of S phase, specifically during the early neuronal fate restrictive differentiation of multipotent neural progenitors. The most profound phenotype that resulted from Nde1 mutations was severe DNA damage that occurred during mid to late S phase heterochromatic DNA replication. Stalled DNA replication led to DNA replicative catastrophe and the activation of tumor suppressor p53 (encoded by Trp53 or Tp53) via DNA damage responses (DDRs). Abrogating p53-suppressed apoptosis rescued the size and structure of Nde1 mutant brain but failed to mitigate the genomic stress. We also identified a nuclear pool of Nde1 and the interaction of Nde1 with the cohesin complex as well as the chromatin remodeler SNF2h. These findings suggest that Nde1 is essential for CNS specific genomic quality control, that chromatin remodeling during heterochromatic replication facilitated by Nde1 is essential for generating cortical layer II/III neurons, and that reduced fidelity of S phase choreography during the early phase of neural progenitor differentiation and identity establishment may lead to mosaic genomic lesions and developmental brain disorders.

## Results

### The loss of Nde1 results in DNA damage in early steps of neuronal fate restriction

The reduction of cortical layer II/III neurons in both Nde1$^{-/-}$ and Nde1$^{-/-}$Lis1$^{+/-}$ mutants was previously found to result from a failure in cell division and precocious neurogenesis of the mutant neural progenitors (*Feng and Walsh, 2004*; *Pawlisz et al., 2008*). However, massive apoptosis detected by TUNEL was also observed predominantly in the newly formed cortical plate (CP) of Nde1$^{-/-}$Lis1$^{+/-}$ embryos at E12.5 (*Pawlisz et al., 2008*). To understand the mechanism of apoptosis and evaluate its contribution to the loss of layer II/III neurons in Nde1 mutant brains, we examined its associated cellular processes and found that the apoptosis in both Nde1$^{-/-}$Lis1$^{+/-}$ and Nde1$^{-/-}$ mutants corresponded with increased DNA damage. A substantial number of cells in the neocortex of Nde1$^{-/-}$Lis1$^{+/-}$ and Nde1$^{-/-}$ mutants showed abnormally high γH2AX immunosignals, a hallmark for DNA double strand breaks (DSBs) (*Thiriet and Hayes, 2005*; *Figure 1A*). While γH2AX foci associated with DNA replication were also widely detected in wild-type and mutant cortices, the γH2AX+ signals observed in the mutant were several orders of magnitude higher, showed pan-nuclear pattern, and were often co-stained by antibodies to cleaved caspase 3 (CC3), the marker for apoptosis. While apoptosis was wide-spread and observed in both CP neurons and ventricular zone (VZ) neural progenitors (*Pawlisz et al., 2008*), γH2AX+ cells were confined in the VZ (*Figure 1A*). This suggested that DNA damage occurred prior to apoptosis and that apoptosis was one of the endpoints of DNA damage caused by Nde1 deficiency.

The DNA damage and apoptosis in Nde1 mutant brain were found to correlate spatiotemporally with the early fate restrictive differentiation of neural progenitors. In the Nde1$^{-/-}$Lis1$^{+/-}$ brain, γH2AX+ and CC3+ cells were detected as early as the onset of neocortical neurogenesis at E10.5 (*Figure 1B*). At this stage, a majority of the neural progenitors progress through the cell cycle rapidly to expand the progenitor pool, while some start to differentiate into intermediate neural progenitors or neurons. From E10.5 to E12.5, as more neural progenitors become fate restricted, both DNA damage and apoptosis in the Nde1 mutant cortex increased and peaked at E12.5. With the progression of cortical neurogenesis, γH2AX+ and CC3+ cells declined gradually after E13.5 when more cortical neurons were produced. Apoptosis in both Nde1$^{-/-}$Lis1$^{+/-}$ and Nde1$^{-/-}$ mutants became lower during the later stages of cortical neurogenesis after E15, even though a majority of cortical layer II/III neurons are

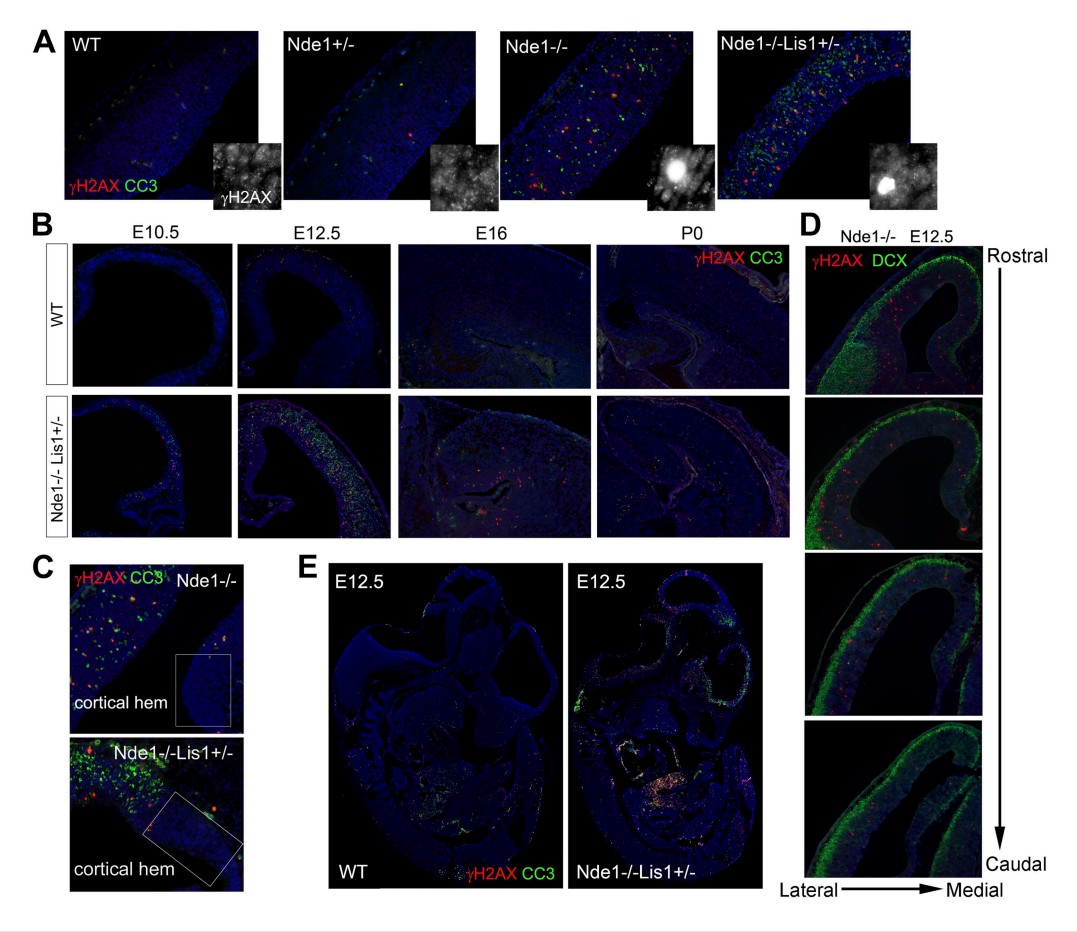

**Figure 1**. The correlation of DNA damage and apoptosis with neural progenitor early fate restriction in Nde1 mutant brains. (**A**) Immunohistological analysis of γH2AX (red) and cleaved caspase 3 (CC3, green) reveals the co-existence of DNA damage and apoptosis in the neocortex of Nde1 mutants at E12.5. Higher-magnification views indicate the high level of γH2AX pan-nuclear signals associated with severe DNA damage and low γH2AX signals associated with normal replication foci. (**B**) Developmental analysis of the temporal correlation between immunosignals of γH2AX (red) and cleaved caspase 3 (CC3, green) from E10.5 to P0. (**C**) The lack of DNA damage and apoptosis in the cortical hem (a region where neural progenitors divide but do not undergo neuronal differentiation) of Nde1$^{-/-}$ and Nde1$^{-/-}$Lis1$^{+/-}$ brains at E12.5. (**D**) Immunohistological analysis of serial coronal sections of Nde1$^{-/-}$ brains to demonstrate the spatial correlation of γH2AX (red) with TNG revealed by DCX abundance (green). (**E**) Immunohistological analysis of γH2AX (red) and cleaved caspase 3 (CC3, green) on sagittal sections of an Nde1$^{-/-}$Lis1$^{+/-}$ embryo and its wild-type littermate at E12.5. Nuclei DNA was stained with Hoechst 33342 and shown in blue in all fluorescent images.

The following figure supplements are available for figure 1:

**Figure supplement 1**. The correlation of DNA damage and apoptosis with neural progenitor early fate restriction in Nde1 mutants.

**Figure supplement 2**. DNA damage and apoptosis along the transverse neurogenetic gradient (TNG) in Nde1$^{-/-}$ brains.

**Figure supplement 3**. The correlation of DNA damage and apoptosis with neuronal differentiation in the Nde1 mutant spinal cord from E9.5 to E13.5.

generated through terminal mitosis between E15 and E17. In contrast to the low level of apoptosis, γH2AX+ cells remained detectable until birth (***Figure 1B***, ***Figure 1—figure supplement 1***). While γH2AX and CC3 signals were at peak levels in the neocortex of Nde1$^{-/-}$Lis1$^{+/-}$ and Nde1$^{-/-}$ brains at

E12.5, they were absent in the mutant cortical hem, the highly proliferating but non-neurogenic hippocampus organizer (*Figure 1C*). This suggested that DNA damage and apoptosis caused by Nde1 mutation were associated with neural progenitor differentiation as opposed to proliferation. During neocortical development, neurogenesis is known to proceed along the transverse neurogenic gradient (TNG), which initiates rostrolaterally and propagates caudomedially (*Caviness et al., 2009*). To better reveal the correlation of DNA damage and apoptosis with neocortical neurogenesis, we examined serially sectioned Nde1$^{-/-}$ brains (n = 6) by double immunostaining of γH2AX and CC3 with newborn neuron markers DCX or Tuj1. Compared to Nde1$^{-/-}$Lis1$^{+/-}$ brains, the well-preserved cytoarchitecture of Nde1$^{-/-}$ brains allows for a better assessment of the spatial distribution of DNA damage and apoptosis with respect to the TNG. At earlier E12, more γH2AX+ cells were observed in the rostrolateral than the caudomedial neocortex. Similarly, CC3+ cells were predominantly detected rostrolaterally in these brains. Towards later E12, as TNG progresses caudomedially, CC3+ cells also increased in the caudolateral and rostromedial cortical regions. In all cortical sections examined, γH2AX+ and CC3 signals were correlated spatially with the presence and abundance of newborn neurons (*Figure 1D*, *Figure 1—figure supplement 2*). Together, these data demonstrated that DNA damage and apoptosis in Nde1 mutant cortices had little correlation with self-renewal proliferation or neurogenic terminal divisions but was well in line with early neural progenitor fate restrictive differentiation around E12.5 in the developing neocortex.

The link of DNA damage and apoptosis with early progenitor differentiation was further shown in the developing spinal cord, where neurogenic domains are better spatiotemporally defined (*Figure 1—figure supplement 3*). We also examined DNA damage and apoptosis on sagittal sections of E12.5 Nde1$^{-/-}$Lis1$^{+/-}$ embryos and found only a small number of cells outside of the developing CNS were affected (*Figure 1E*). These developmental analyses demonstrate that the most profound phenotype caused by Nde1 mutations is characterized by DNA damage and apoptosis, and that this phenotype is spatiotemporally in parallel to the early differentiation period of neural progenitors. Thus, the indispensible role of Nde1 in generating layer II/III neurons appears to be at the time when the fate of these neurons is initially established rather than when they are generated through terminal mitosis.

## Apoptosis resulted from DNA damage response

To confirm that the apoptosis in Nde1 mutants was the result of DNA damage but not vice versa, we detected DNA damage directly by performing the comet assay on primary cortical cells (*Collins, 2004*; *Olive and Banath, 2006*). Compared to cells from Nde1$^{+/-}$ embryos, more Nde1$^{-/-}$Lis1$^{+/-}$ and Nde1$^{-/-}$ cells showed longer comet tails, indicating that loss of Nde1 function indeed resulted in an increase in DNA lesions in the developing cortex (*Figure 2A*).

DNA damage is known to initiate DNA damage responses (DDRs) primarily through the activation of ATM/ATR kinases, which in turn directly phosphorylate p53 on Ser18 (Ser15 in human) (*Banin et al., 1998*; *Canman et al., 1998*; *Khanna et al., 1998*) in addition to H2AX. Double immunostaining of γH2AX and phospho p53 at Ser18 showed their co-activation in mutant neocortical neural progenitors (*Figure 2B,C*). Many mutant cells showed elevated γH2AX and p-p53S18 but lacked DNA condensation and fragmentation, supporting that γH2AX and p53 activation in Nde1 mutants occurred prior to the programmed cell death (*Figure 2C*). We found that the level of p53 S18 hyper-phosphorylation correlated to Nde1 gene dosage as well as to the degree of brain malformations caused by Nde1 mutations (*Figure 2D*). Similarly, the loss of Nde1 resulted in an elevated basal level of p53, which is known to be low in healthy cells, supporting the DDR-mediated p53 activation. These results indicate that Nde1 mutation primarily results in DNA damage, and apoptosis is caused by DDR with p53 activation. Thus, in contrast to the prevailing view of Nde1 being a cytoskeletal regulator, these data revealed a novel function of Nde1 in protecting the genome.

## p53-dependent apoptosis in Nde1 mutant brain

To further establish Nde1's essential role in safeguarding the genome during cortical neural progenitor differentiation, we abrogated p53 in Nde1-deficient mice. As expected, the loss of p53 abolished apoptosis in Nde1$^{-/-}$ progenitors and restored the size of the Nde1 mutant brain (*Figure 3A,B*, *Figure 3—figure supplement 1A*). The brain of Nde1$^{-/-}$Trp53$^{-/-}$ double mutant mice was found to be nearly the same as those of wild-type in both size and structure at the weaning age. Histological and immunohistological analyses showed all anatomical features, especially the thickness of layer II/III neurons was fully restored (*Figure 3C,D*). Quantitative analysis of the fraction of Cux1 immunolabeled upper layer

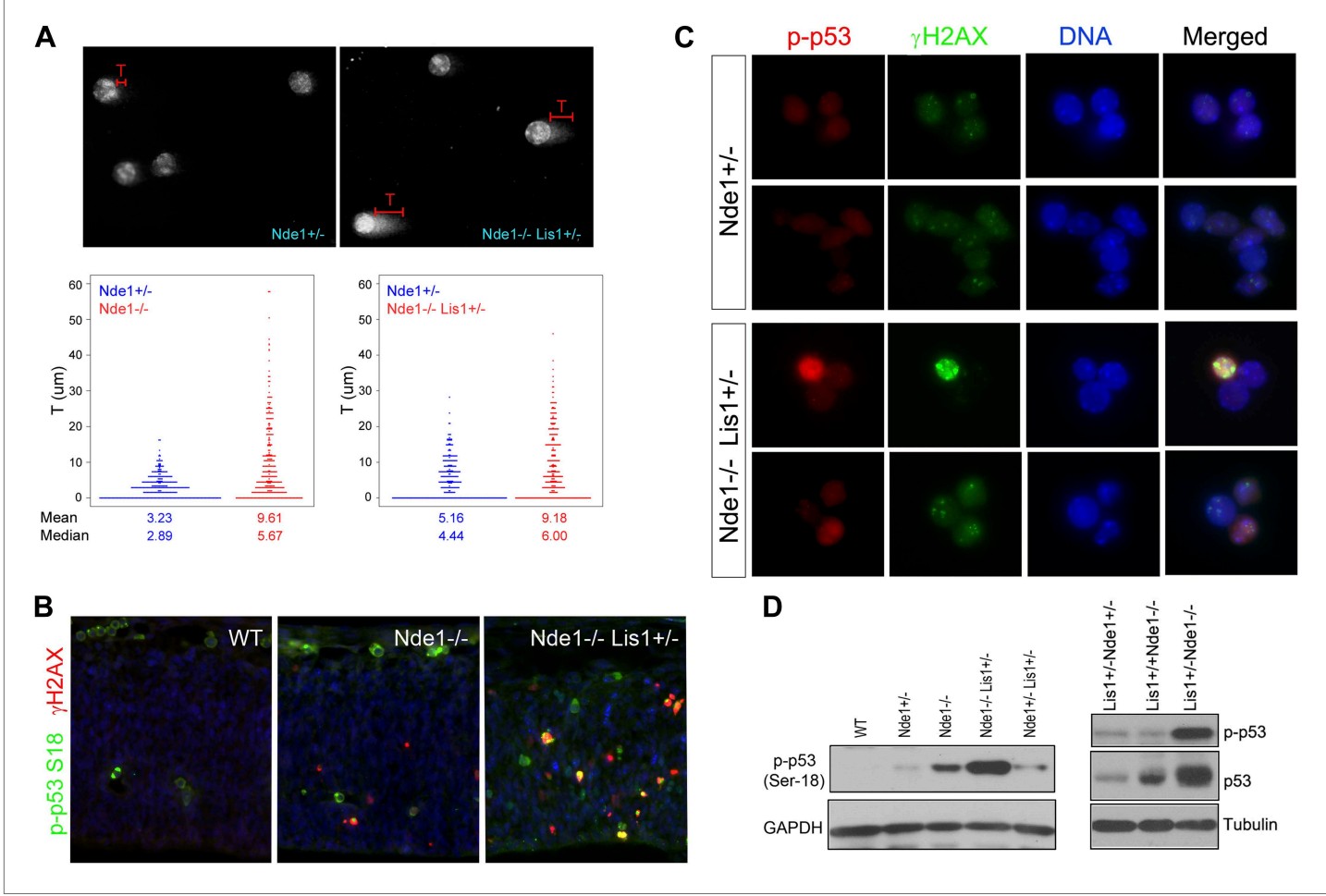

**Figure 2**. Co-activation of γH2AX with p53 in Nde1 mutant neocortices. (**A**) Results from comet assay with cortical cells isolated from Nde1$^{-/-}$ or Nde1$^{-/-}$Lis1$^{+/-}$ mutants and their Nde1$^{+/-}$ littermates at E12.5. The distribution, mean, and median values of comet tail length (T) from over 300 randomly selected and photographed cells are presented. Nde1$^{-/-}$ and Nde1$^{-/-}$Lis1$^{+/-}$ cells showed increased comet tail length compared to Nde1$^{+/-}$ cells, respectively; p < 0.0001 by the Wilcoxon rank-sum two sample test. (**B**) Immunohistological analysis of the γH2AX (red) and phospho-p53 Ser18 (p-p53, green) in the neocortex of wild-type and Nde1 mutants at E12.5. (**C**) Immunofluorescence analysis of the co-activation of γH2AX (green) and phospho-p53 Ser18 (red) in primary cortical cells isolated at E12.5. Note the lack of DNA condensation and fragmentation of cells with high phospho-H2AX and p53 signals. (**D**) Immunoblotting analyses of phospho-p53 Ser18 in embryonic cortical lysates at E12.5. Nuclei DNA was stained with Hoechst 33342 and shown in blue in all fluorescent images.

projection neurons (with respect to total NeuN+ or NeuN+Cux1-neurons) in the Nde1$^{-/-}$Trp53$^{-/-}$ double mutant brain indicated that they were at the wild-type-level (*Figure 3E*, *Figure 3—figure supplement 1B,C*). These results demonstrate that microcephaly and reduced layer II/III neurons caused by the loss of Nde1 is a result of p53-dependent apoptosis elicited by DDR and that NDE1 is specifically essential for the genomic integrity of neurons that conduct higher order brain functions.

## The association of DNA damage with DNA replication

Consistent with an essential role of Nde1 in early neuronal fate restriction, we found that active DSBs and DDR in the Nde1$^{-/-}$ brain occurred in the primary multipotent neural progenitors in the VZ but was rarely detected in the intermediate neural progenitors of the sub-ventricular zone (SVZ), and not in the post-mitotic neurons in the CP (*Figure 4A*, *Figure 4—figure supplement 1A,B*). To identify the source of DSBs caused by Nde1 mutations, we examined their occurrence with respect to the cell cycle. After BrdU transient labeling, we found about 20% of γH2AX+ progenitors were also BrdU+ (*Figure 4B*). In contrast, co-immunostaining of γH2AX and phospho-histone H3 (PH3) failed to show cells with elevated γH2AX during mitosis (*Figure 4—figure supplement 1C*). Because of the transient nature of

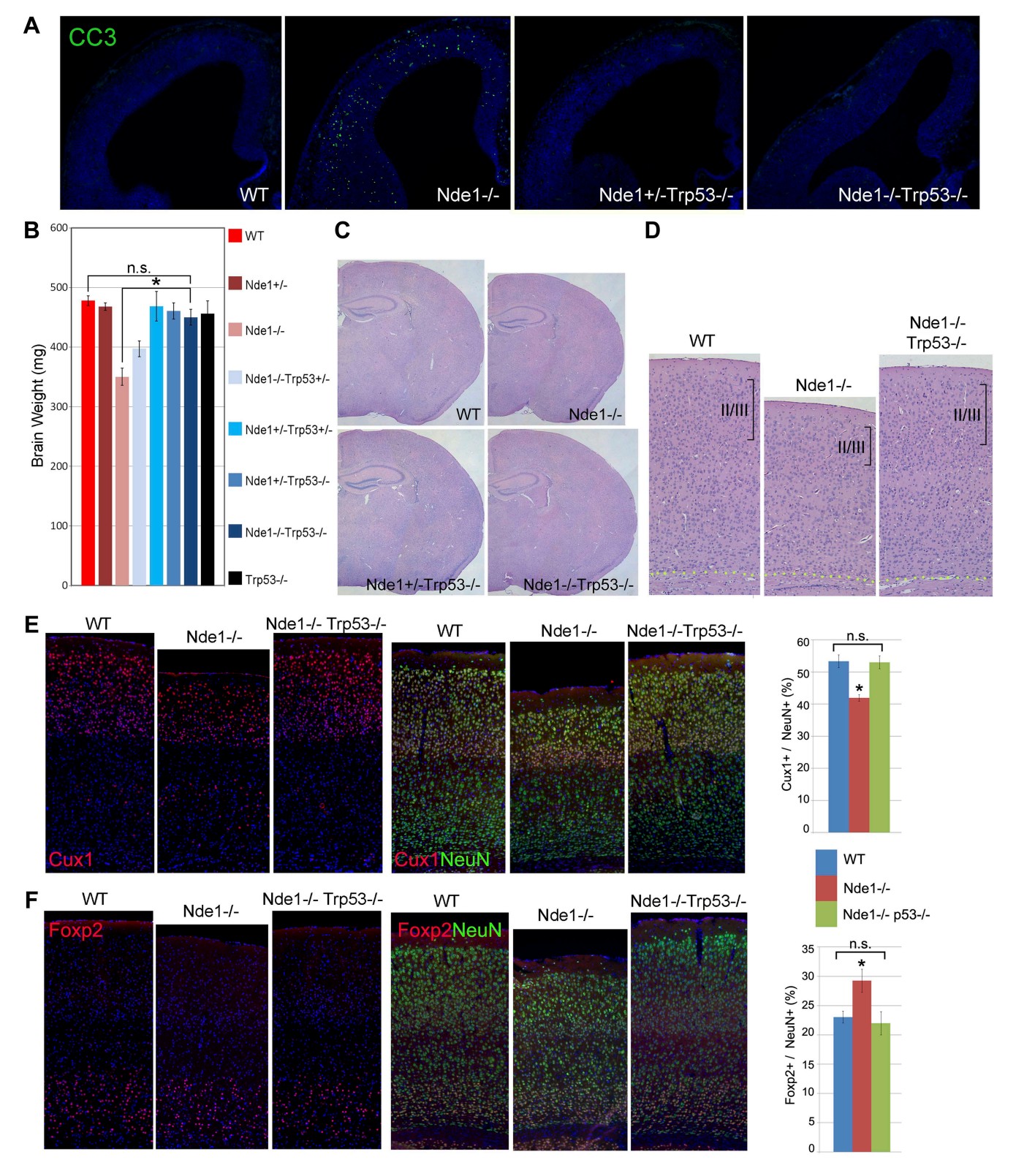

**Figure 3**. Restoration of the size and structure of the Nde1$^{-/-}$ brain by abrogating p53. (**A**) Immunohistological analyses of cleaved caspase 3 (CC3, green) in wild-type, Nde1$^{-/-}$, Nde1$^{+/-}$Trp53$^{-/-}$, and Nde1$^{-/-}$Trp53$^{-/-}$ neocortices at E12.5. (**B**) Brain weight of Nde1$^{-/-}$, Nde1$^{-/-}$Trp53$^{-/-}$ mutant mice and their littermates at post-natal for 3 to 4 weeks. Data are mean ± SD. Significant overall differences were found among wild-type and Nde1–Trp53 double
*Figure 3. Continued on next page*

*Figure 3. Continued*

mutants by ANOVA (p < 0.0001). Pairwise comparisons showed that the brain mass of Nde1$^{-/-}$Trp53$^{-/-}$ mice (n = 8) was significantly increased compared to that of Nde1$^{-/-}$ mice (*p < 0.0001), but not significantly different from that of the wild-type mice (n.s., p = 0.06). (**C**) H&E stained brain sections of wild-type, Nde1$^{-/-}$, Nde1$^{+/-}$Trp53$^{-/-}$, and Nde1$^{-/-}$Trp53$^{-/-}$ mice reveal normal size and anatomical structure of the Nde1$^{-/-}$Trp53$^{-/-}$ brain. (**D**) H&E stained cortical sections of wild-type, Nde1$^{-/-}$, and Nde1$^{-/-}$Trp53$^{-/-}$ brains, showing restored layer II/III cortical neurons in the Nde1$^{-/-}$Trp53$^{-/-}$ brains. (**E**) Immunohistological and quantitative analyses of the number and distribution of Cux1+ (red) superficial layer cortical neurons. Cortical neurons were identified by NeuN immunoreactivities (green); nuclei DNA was stained with Hoechst 33342 and shown in blue. Data are presented as mean ± SD in percentage of total NeuN+ neurons (n = 5). Significant overall differences were found among wild-type, Nde1$^{-/-}$, and Nde1$^{-/-}$Trp53$^{-/-}$ brains by ANOVA (p < 0.0001). Pairwise comparisons indicated that compared to the wild-type, Cux1+ neurons were significantly decreased in the Nde1$^{-/-}$ brains (*p < 0.0001), but not significantly changed in Nde1$^{-/-}$Trp53$^{-/-}$ brains (n.s., p = 0.94). Compared to Nde1$^{-/-}$, Cux1+ neurons were significantly increased in Nde1$^{-/-}$Trp53$^{-/-}$ brains (p < 0.0001). (**F**) Immunohistological and quantitative analyses of the number and distribution of Foxp2+ (red) deep layer cortical neurons. Significant overall differences were found among wild-type, Nde1$^{-/-}$, and Nde1$^{-/-}$Trp53$^{-/-}$ brains by ANOVA (p < 0.0001). Pairwise comparisons indicated that compared to the wild-type, Foxp2+ neurons were significantly increased in the Nde1$^{-/-}$ brains (*p<0.005), but not significantly changed in Nde1$^{-/-}$Trp53$^{-/-}$ brains (n.s., p = 0.77). Compared to Nde1$^{-/-}$, Foxp2+ neurons were significantly decreased in Nde1$^{-/-}$Trp53$^{-/-}$ brains (p = 0.001).

The following figure supplement is available for figure 3:

**Figure supplement 1**. Restoration of brain size and structure of Nde1$^{-/-}$ mutants by abrogating p53.

both γH2AX and the BrdU pulse, these results demonstrate that DNA damage caused by Nde1 deficiency can occur during DNA replication.

The concurrence of DNA damage and DNA replication in Nde1 mutant progenitors suggested an essential function of Nde1 in S phase. To confirm this, we measured the duration of S phase (Ts) of primary progenitors by sequential IdU-BrdU labeling and analysis (*Martynoga et al., 2005*). In this experiment, IdU was used to label progenitors in S phase, and BrdU was added 1.5 hr later to assess if IdU+ cells had completed DNA replication. Our data showed that the Ts of Nde1$^{-/-}$ progenitors was significantly longer than that of the wild-type cells (*Figure 4C*). Furthermore, we also noticed that in both Nde1$^{-/-}$Lis1$^{+/-}$ and Nde1$^{-/-}$ mutants, many nuclei that stopped incorporating BrdU (IdU+BrdU−) remained in the S phase zone of the basal VZ instead of descending apically as seen in the wild-type progenitors (*Figure 4D*, white arrows). Neural progenitors in the cortical VZ are polarized apical-basally and known to undergo interkinetic nuclear migration (INM) by moving their nuclei basally during S phase. After S phase is completed, the nuclei are moved apically for mitosis to occur at the ventricular surface (*Takahashi et al., 1993*). As the basally localized IdU+BrdU− nuclei in the mutant were not recognized in G2/M by PH3 (*Figure 4—figure supplement 2A*), it suggested that these mutant cells might arrest in the basal S phase zone due to the stalled DNA replication. To confirm the faulty S phase prior to G2/M, we increased the time between IdU and BrdU pulses to 2.5 hr to observe if the IdU+BrdU− cells in the basal S phase zone of the mutant may progress to mitosis over time. However, we observed that rather than progressing to mitosis, these cells remained stalled and that about half (47.5%) of the γH2AX+ cells were positively stained by antibodies to IdU but not to BrdU, indicating that S phase-stalled cells underwent DDR by activating γH2AX (*Figure 4E*, *Figure 4—figure supplement 2B*). This explains why only 20% of γH2AX+ cells were actively incorporating BrdU (*Figure 4B*), since a majority of γH2AX+ cells were stalled in the S phase. Therefore, our data demonstrate that Nde1 is functionally essential during S phase and that DNA damage in Nde1 mutant progenitors results from stalled and catastrophic DNA replication.

## S phase impairment during heterochromatic replication

During the S phase of metazoan cells, various chromosomal domains are known to replicate in a well-defined spatiotemporal sequence and can be visualized experimentally by labeling and immunostaining with nucleotide analogs (*Dimitrova and Gilbert, 1999*; *Dimitrova et al., 2002*; *Maison et al., 2010*). To better understand the function of Nde1 in S phase, we studied the dynamic progression of S phase and analyzed IdU and CldU sequentially labeled chromosomal domains by taking advantage of the non-overlapping immunosignals between IdU and CldU. Embryos were pulse labeled by IdU for 1.5 hr before being labeled again by CldU transiently. Cells in early S phase were identified by CldU but not IdU incorporation (IdU−CldU+) as well as a more evenly distributed CldU immunosignals, as new replication foci emerge continuously throughout the transcriptionally active euchromatin. Whereas cells in mid to late S phase were labeled by both IdU and CldU (IdU+CldU+) and recognized by

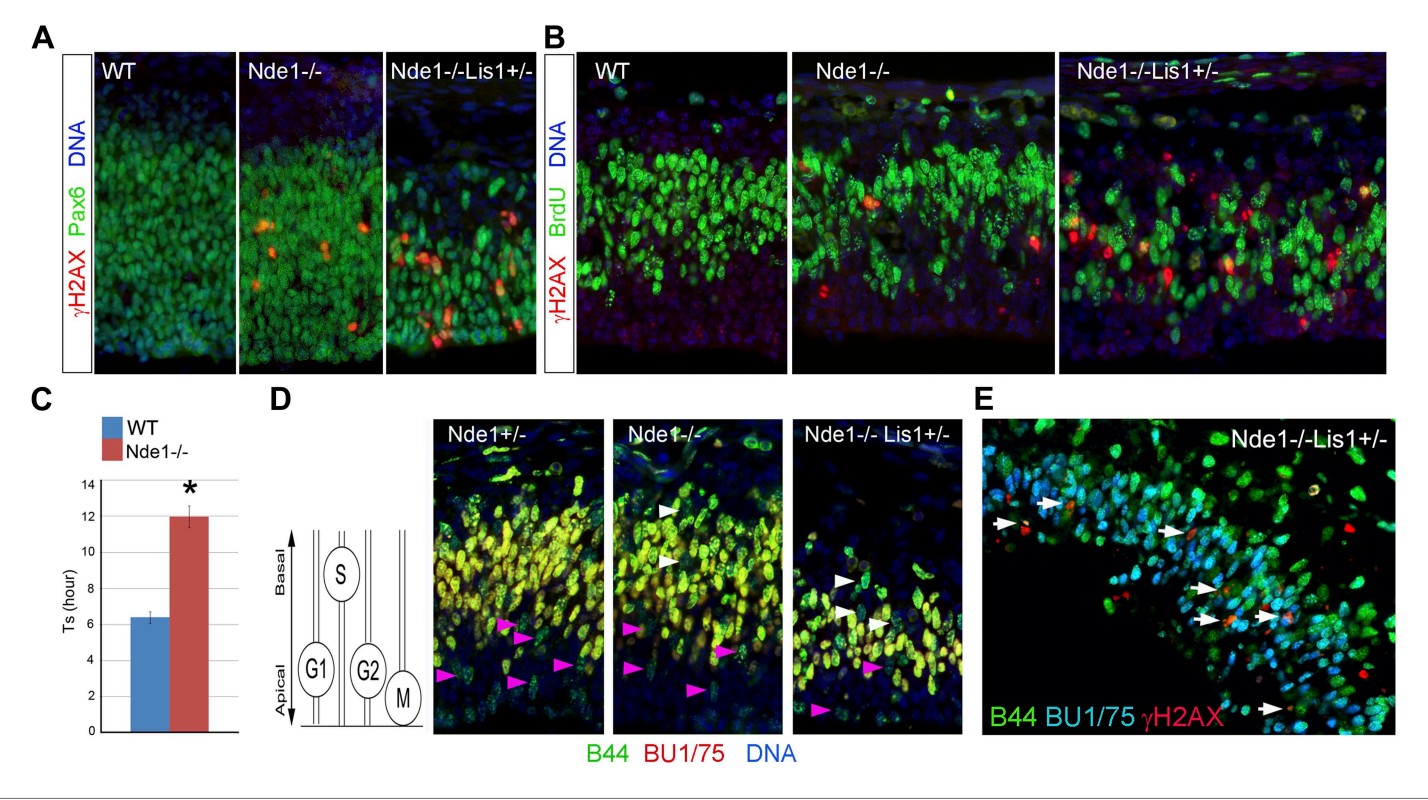

**Figure 4**. DNA damage caused by Nde1 mutation occurs concurrently with DNA replication. (**A**) Co-immunostaining of γH2AX (red) and multipotent/primary progenitor marker Pax6 (green) on cortical sections at E12.5. (**B**) Co-immunostaining of γH2AX (red) and BrdU (green); cortical sections were prepared from BrdU pulse (30 min) labeled embryos at E12.5. Quantification of γH2AX+ cells showed that 18.3 ± 6.1% (mean ± SD) of γH2AX+ Nde1$^{-/-}$ and 23.4 ± 3.1% (mean ± SD) of γH2AX+ Nde1$^{-/-}$Lis1$^{+/-}$ cells were also BrdU+. (**C**) S phase duration (Ts, hr) measurement indicates significant delay of DNA replication in Nde1$^{-/-}$ neural progenitors. Data are mean ± SD, p < 0.0001 by Student's t test. (**D**) Representative images of B44 (green, recognizes both IdU and BrdU) and BU1/75 (red, recognizes only BrdU) double immunohistological staining in an IdU (2 hr), BrdU (30 min) sequential labeling experiment. Cells that have finished S phase and progressed to G2/M are indicated by pink arrows; cells that remained in the S phase zone but stopped incorporating BrdU are indicated by white arrows. A diagram to indicate the cell cycle dependent nuclei position through INM is included. (**E**) B44 (green), BU1/75 (blue), and γH2AX (red) triple immunostaining of cortical sections of Nde1$^{-/-}$Lis1$^{+/-}$ mutant after IdU (2.5 hr), BrdU (30 min) sequential labeling. 47.5 ± 0.1% (mean ± SD) of total γH2AX+ cells were B44+BU1/75−, indicating the association of DNA damage with stalled DNA replication (white arrows).

The following figure supplements are available for figure 4:

**Figure supplement 1**. The cell type and cell cycle specificity of DNA damage in Nde1 mutant brains.

**Figure supplement 2**. Stalled or delayed DNA replication in Nde1 mutant neural progenitors.

enhanced nucleotide incorporation in various transcriptionally silent heterochromatic domains. Cells that had left S phase could be noted by IdU+ only (IdU+CldU−) immunosignals, since they stopped nucleotide incorporation before CldU pulse labeling. By quantitative comparison of IdU-CldU labeled progenitors in the cortical VZ of wild-type and Nde1$^{-/-}$ mutant, we failed to detect a significant difference in the early S phase population but found more mutant progenitors in mid to late S phase (*Figure 5A*). In a large number of Nde1$^{-/-}$ progenitors, CldU signals in IdU+CldU+ cells were predominantly detected along the nuclear periphery, the rim of nucleoli, and on particles of satellite repeats which are characteristic for heterochromatic domains, indicating replication difficulties in these domains during mid to late S phase (*Figure 5B–D*). Triple immunostaining of IdU, CldU, and PH3 allowed us to access the S to G2/M progression; we found that Nde1$^{-/-}$ progenitors showed a reduction in the intensity and the number of IdU labeled foci in PH3+ cells (*Figure 5B*, arrows), suggesting reduced nucleotide incorporation into mutant progenitors before S phase completion. Collectively, the unaltered

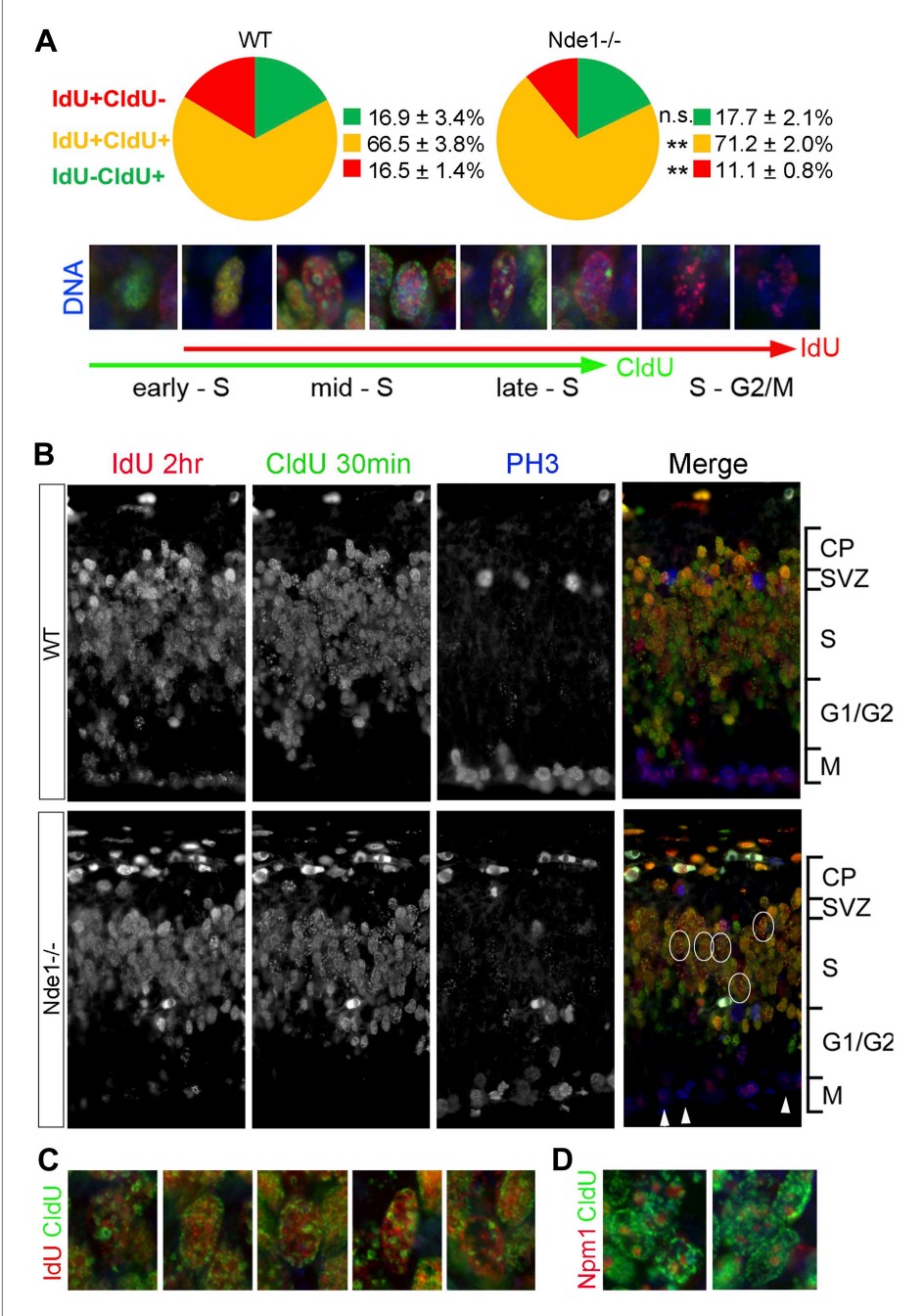

**Figure 5**. Stalled DNA replication during mid-late S phase at heterochromatic domains in Nde1$^{-/-}$ mutant neural progenitors. (**A**) Quantitative analysis of IdU+CldU− (red), IdU+CldU+ (yellow), and IdU−CldU+ (green) cell fractions (%) by IdU (2 hr) and CldU (30 min) sequential labeling. Data are presented as mean ± SD. n.s.: p > 0.05; **: p < 0.001 by Chi–Square tests. A diagram that shows spatiotemporal patterns of early, mid, and late S phase DNA replication is included. (**B**) Representative images of B44 (red), BU1/75 (green), and PH3 (blue) triple immunostaining from IdU (2 hr) and CldU (30 min) sequential labeling experiments. Note the CldU (BU1/75, green) signals in IdU+CldU+ progenitors highlight predominantly heterochromatic structures (circled cells, better revealed in **C**) and the low IdU (B44, red) signals in PH3+ cells of the Nde1$^{-/-}$ mutant (arrows). (**C**) Higher-magnification views of selected Nde1$^{-/-}$ mutants progenitors (circled in **B**) that show DNA replication at heterochromatin (nuclear periphery, the rim of the nucleoli, and large foci of repeated heterochromatic sequences). (**D**) Double immunohistological stain of CldU (green) and nucleolar marker Nucleophosmin 1 (Npm1, red) to view stalled DNA replication at perinuclear heterochromatin, which is known to comprise centromeres and pericentromeres.

early S phase population, significant increase in CldU labeling in various heterochromatic domains, reduced IdU labeling in PH3+ cells, combined with prolonged duration of S phase, suggest that DNA replication in Nde1$^{-/-}$ progenitors progresses normally through the euchromatin, but stalls during heterochromatin replication in mid-late S phase.

## p53 abrogation fails to rescue genome damage

The high level of p53-dependent apoptosis in Nde1$^{-/-}$ brain suggested that the DNA replication catastrophe activated DDR to eliminate neural progenitors or neurons with genomic lesions. In this case, the restored size and structure of Nde1$^{-/-}$ brain by p53 abrogation is not expected to show genome rescue even though DDR-induced apoptosis was suppressed. This would also suggest that genomic lesions incurred during development could be structurally tolerated and not necessarily manifest in changed brain anatomy. To test this hypothesis, we measured the Ts of Nde1$^{-/-}$Trp53$^{-/-}$ progenitors and found the lengthened Ts in Nde1$^{-/-}$ progenitors was not mitigated by p53 removal (*Figure 6A*). Likewise, the IdU-CldU sequential labeling study also indicated delayed DNA replication at heterochromatic domains in Nde1$^{-/-}$Trp53$^{-/-}$ neural progenitors (*Figure 6B*). Reduced BrdU or IdU incorporation was found when Nde1$^{-/-}$Trp53$^{-/-}$ progenitors progressed from S to G2/M in 1.5 hr, which was in agreement with hindered nucleotide incorporation before S phase completion (*Figure 6B,C*, arrows). These observations suggest that replication stress and genomic impairments remained with the absence of apoptosis.

The lack of apoptosis as well as the restored cortical structure and number of layer II/III neurons in the Nde1$^{-/-}$Trp53$^{-/-}$ brain did not appear to associate with a fully rescued genome. As somatically generated chromosomal aberrations can happen in neocortical progenitors and neurons of the normal brain (*Rehen et al., 2001*; *Bushman and Chun, 2013*), and Nde1 mutation affects only selected cells in a heterogeneous pool of neural progenitors, such genomic mosaicism and cell heterogeneity of the developing cortex precluded us from obtaining direct information on the genomic lesions in the Nde1$^{-/-}$Trp53$^{-/-}$ brain. Instead, we looked into further evidence of DNA damage by examining cell cycle stress in Nde1$^{-/-}$Trp53$^{-/-}$ embryonic cortices. After completing S phase, neural progenitors of the VZ normally move replicated chromosomes through INM towards the apical ventricular surface where chromosome condensation and mitosis occur. In both Nde1$^{-/-}$ and Nde1$^{-/-}$Trp53$^{-/-}$ cortices, a substantial number of PH3+ progenitors were found mislocalized basally (*Figure 6C,D*). PH3 is an indicator of cells in both late-G2 and M phases; we found chromosomes of the mislocalized PH3+ mutant progenitors were not condensed (*Figure 6C,D* higher magnification views), indicating cell cycle arrest in G2 instead of M phase. The G2 arrest in Nde1$^{-/-}$ and Nde1$^{-/-}$Trp53$^{-/-}$ progenitors was confirmed by co-immunostaining PH3+ cells with the phospho-vimentin antibody 4A4, which exclusively recognizes progenitors in M phase (*Weissman et al., 2003*). Most PH3+4A4+ progenitors in Nde1$^{-/-}$ and Nde1$^{-/-}$Trp53$^{-/-}$ mutants were correctly localized at the ventricular surface, whereas most PH3+4A4− progenitors in the mutants were ectopically localized, indicating that the basally mislocalized PH3+ cells were those arrested in late-G2 and that INM in the mutant was largely normal (*Figure 6D*). Consistent with the G2 arrest, we found that the level of PH3 was elevated in the Nde1$^{-/-}$Trp53$^{-/-}$ brain (*Figure 6E*). The hallmark for DNA damage-induced G2/M checkpoint involves the degradation of Cdc25A, a phosphatase essential for Cdk1 activation (*Mailand et al., 2000*). Both Nde1$^{-/-}$ and Nde1$^{-/-}$Trp53$^{-/-}$ brains showed reduced levels of Cdc25A at E12.5 (*Figure 6E*), supporting the notion that prolonged G2 in Nde1$^{-/-}$Trp53$^{-/-}$ progenitors was a result of persistent DNA damage.

We followed a cohort of Nde1 Trp53 double mutant mice (n > 80) over a 4-month period, and none of the Nde1$^{-/-}$Trp53$^{-/-}$ mice (n > 20) was found to have altered brain structures or tumors of CNS origin, though they showed obvious tumors in multiple organs outside of the CNS. We did not find tumors in Nde1$^{+/-}$Trp53$^{+/-}$ mice (n > 30), but approximately one third of the Nde1$^{-/-}$Trp53$^{+/-}$ mice (~10 out of >30) developed tumors predominantly of hematopoietic lineages including lymphoma and tumor of the thymus (*Figure 6—figure supplement 1A*). Apoptosis and mitotic figures along with remarkably elevated p53 protein levels were observed in the Nde1$^{-/-}$Trp53$^{+/-}$ thymus tumor, suggesting malignancy and increased stress in the tumor tissue (*Figure 6—figure supplement 1B,C*). The tumor formation in the Nde1$^{-/-}$Trp53$^{+/-}$ mice is in agreement with Nde1's selective expression and functional requirement in hematopoietic cells in addition to neural progenitors (*Figure 6—figure supplement 1D*). Together, the experimental evidence also supported that although p53 abrogation could restore the size and number of cortical layer II/III neurons of Nde1$^{-/-}$ brain, it unlikely ameliorated the genomic lesion caused by the Nde1 deficiency.

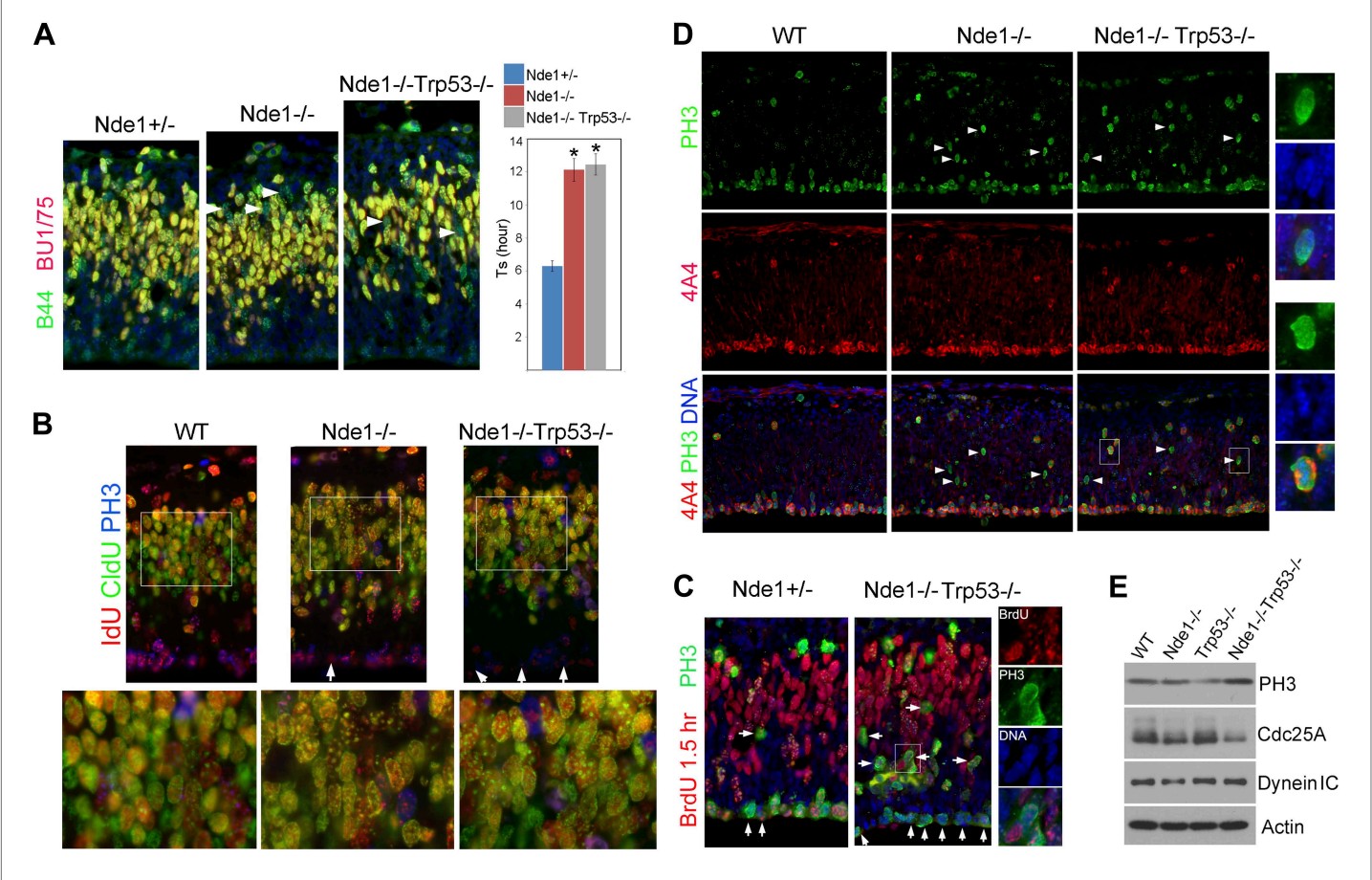

**Figure 6**. Persistent cell cycle stress and genotoxicity in Nde1 mutants after p53 abrogation. (**A**) Representative images of B44 (green, recognizes both IdU and BrdU) and BU1/75 (red, recognizes only BrdU) double immunostained neocortical sections from IdU (2 hr), BrdU (30 min) sequential labeling experiments to measure S phase duration (Ts). Cells that were stalled in the S phase zone (IdU+ BrdU−; green) are indicated by white arrows. Significant overall difference in Ts was found among wild-type, Nde1$^{-/-}$, and Nde1$^{-/-}$Trp53$^{-/-}$ progenitors by ANOVA (p < 0.0001). Pairwise comparisons indicated that Ts of Nde1$^{-/-}$Trp53$^{-/-}$ progenitors was significantly longer than that of wild-type progenitors (*p < 0.0001) but not significantly different from that of the Nde1$^{-/-}$ progenitors (n.s., p = 0.74). Data are presented as mean ± SD. (**B**) Representative images of B44/IdU (red), BU1/75/CldU (green), and PH3 (blue) triple immunostaining of neocortical sections from embryos sequentially labeled by IdU (2 hr) and CldU (30 min). Note the enhanced CldU immunosignals at heterochromatin structures (higher magnification views) and reduced IdU immunosignals in PH3+ cells (arrows) in Nde1$^{-/-}$ and Nde1$^{-/-}$Trp53$^{-/-}$ cortical neural progenitors. (**C**) Immunohistological analysis of BrdU (red)–PH3 (green) co-labeled cells 1.5 hr after BrdU pulse. Arrows indicate PH3+ cells with very few BrdU foci, suggesting hindered BrdU incorporation at the end of S phase. Higher magnification views of a BrdU–PH3 double positive Nde1$^{-/-}$Trp53$^{-/-}$ cell with uncondensed DNA are also shown. (**D**) Double immunostaining with G2/M marker PH3 (green) and M phase marker phospho-vimentin 4A4 (red) showing increased PH3+4A4− G2 population in Nde1$^{-/-}$ and Nde1$^{-/-}$Trp53$^{-/-}$ progenitors (arrows). Higher magnification views of selected Nde1$^{-/-}$Trp53$^{-/-}$ cells are included to show uncondensed DNA in PH3+4A4− but condensed DNA in PH3+4A4+ cells. (**E**) Immunoblotting analysis of neocortical lysates showing elevated PH3 and increased Cdc25A degradation in the Nde1$^{-/-}$Trp53$^{-/-}$ mutant brain. β-actin (Actin) and dynein intermediate chain (Dynein IC) were used as loading controls.

The following figure supplement is available for figure 6:

**Figure supplement 1**. Lymphomagenesis and remarkably increased p53 in Nde1$^{-/-}$Trp53$^{+/-}$ thymic lymphoma.

## Interaction of Nde1 with cohesin and SNf2h

To understand the molecular mechanism by which Nde1 safeguards the heterochromatin replication, we re-examined its subcellular localization and molecular interactions. Nde1 is a dynamic scaffold regulated by cell cycle dependent post-translational modifications, and it is capable of participating in various cellular compartments (*Feng and Walsh, 2004*; *Stehman et al., 2007*; *Pawlisz and Feng, 2011*). By varying conditions of immunodetection, we found Nde1 was localized in the nucleus of

a subset of primary progenitors identified by Pax6 immunoreactivity in the neocortical VZ at E12.5 (*Figure 7A*). The possible nuclear targeting of Nde1 was confirmed by transfecting and examining GFP-tagged Nde1 in HeLa cells using confocal microscopy (*Figure 7B*). Furthermore, co-immunostaining of Nde1 and Pcna, a DNA polymerase accessory factor which marks the replication fork with a distinctive punctate pattern, indicated that the nuclear pool of Nde1 was elevated when the progenitors were undergoing DNA replication (*Figure 7C*), which suggested that Nde1 may translocate into the nucleus during S phase to play a direct role in DNA replication.

To further assess the involvement of Nde1 in DNA replication, we searched for Nde1 binding proteins and identified structural maintenance of chromosomes 3 (Smc3) as a novel Nde1 partner from an Nde1 yeast two-hybrid screen. Smc3 is a core subunit of cohesin, an essential guardian of genome integrity. Composed of SMC1, SMC3, SCC1 (RAD21), and SCC3 (STAG), cohesin forms a ring-like structure to embrace chromatin fibers from the moment they arise following the replication fork until the onset of anaphase to ensure the proper segregation of genetic materials to daughter cells (*Nasmyth, 2011*). The physical interaction of Nde1 and Smc3 was confirmed by the co-immunoprecipiation studies (*Figure 7D*). Two other core subunits of the cohesin complex, Smc1 and Rad21, were also detected in the Nde1–Smc3 immunocomplex, indicating the direct interaction of Nde1 with the entire cohesin complex. We also performed the reciprocal co-immunoprecipitation analysis to detect Nde1 in the immunoprecipitate of the SMC3 fragment recovered from the Nde1 yeast two-hybrid screen. This Nde1 binding domain of Smc3 (NBD), which corresponds to amino acids R746 to R903 in the coiled-coil region between the C-terminal globular and central hinge domains of Smc3, was immunoprecipitated and found to form a specific immunocomplex with Nde1 (*Figure 7E*). To determine the functional significance of the Nde1–Smc3 interaction, we overexpressed the NBD to block the binding of Nde1 with cohesin and found that NBD arrested the cell cycle preferentially in S phase (*Figure 7F*). We also noted that many cells expressing the NBD showed fragmented nuclei or committed to apoptosis as seen in Nde1 mutant neural progenitors (*Figure 7—figure supplement 1*). These data suggested that the lack of proper interaction of Nde1 with cohesin may underlie the S phase genotoxicity in Nde1 mutant neural progenitors.

Besides a canonical role in maintaining the fidelity of chromosome segregation, cohesin has emerged as an important regulator for higher order chromatin structures, essential for DNA replication through densely packed heterochromatic repetitive sequences. Cohesin can also serve as a platform for DSB repair or act as a nuclear global controller for gene expression on both transcriptional and epigenetic levels. The diverse functions of cohesin are accomplished through recruiting various molecules to the core cohesin ring complex (*Peters et al., 2008*; *Imakaev et al., 2012*; *Baranello et al., 2014*). To define the functional cooperation of Nde1 with cohesin, we tested if Nde1 co-complexes with cohesin partners for DNA repair, chromatin remodeling, and transcriptional regulation through co-immunoprecipitation studies. The chromatin remodeler SNF2h was consistently found in the Nde1–cohesin immunoprecipitates (*Figure 7G*). SNF2h is an ATPase chromatin remodeler that interacts with RAD21 (*Hakimi et al., 2002*). During S phase, SNF2h is recruited to the replication foci and facilitates heterochromatic remodeling (*Zhou et al., 2009*; *Guetg et al., 2010*; *Sugimoto et al., 2011*; *Postepska-Igielska et al., 2013*). The participation of Nde1 in the cohesin–SNF2h complex suggests that Nde1 may participate in mid to late S phase heterochromatin remodeling.

Karyotyping analysis of mouse embryonic fibroblast (MEFs) also revealed that Nde1 mutation led to increased genomic instability similar to those that have been seen in cohesinopathy, and that Nde1 Trp53 double mutation accelerated the aneuploidy caused by the Nde1$^{-/-}$ single mutation. At passage 3, when Nde1$^{-/-}$ MEFs only showed a modest increase in aneuploidy compared to the wild-type MEFs, more than 50% of the Nde1$^{-/-}$Trp53$^{-/-}$ MEFs were aneuploid. About 70% of the Nde1$^{-/-}$ MEFs became aneuploid by passage 10, but a similar degree of aneuploidy developed in the Nde1$^{-/-}$Trp53$^{-/-}$ MEFs at passage 5 (*Figure 7—figure supplement 2A*). While some Nde1$^{-/-}$ cells showed chromosomal aberrations with visible chromosome breaks on crossed chromosome arms, which was reminiscent of what have been observed in cohesin mutants (*Peters et al., 2008*), more severe chromosomal aberrations including breaks, fragmentation, detachments, and cross of chromosome arms were frequently observed in Nde1$^{-/-}$Trp53$^{-/-}$ MEFs (*Figure 7—figure supplement 2B,C*). The increased chromosomal instability of Nde1$^{-/-}$Trp53$^{-/-}$ over Nde1$^{-/-}$ MEFs was also in agreement with what was observed in Nde1$^{-/-}$Trp53$^{-/-}$ neural progenitors (*Figure 6*) and supported that abrogating p53 increased genotoxic lesions in the Nde1$^{-/-}$ mutant by allowing mutant cells with DSBs to escape DDR-induced p53-dependent apoptosis, enabling the genetically altered mutant cells to survive and propagate.

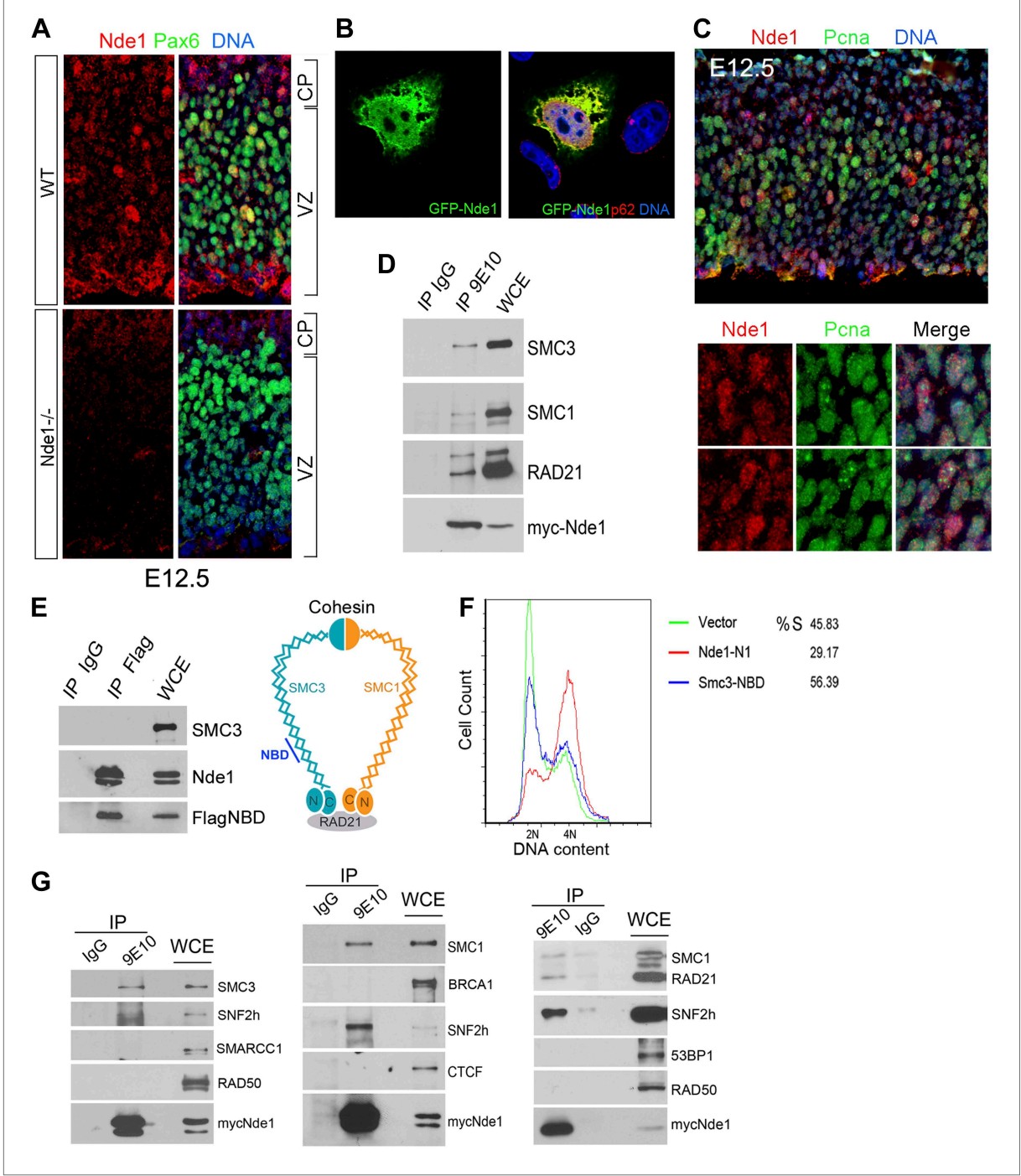

**Figure 7**. Identification of a nuclear pool of Nde1 that interacts with the cohesin complex. (**A**) Double immunohistological staining with antibodies to Nde1 (red) and Pax6 (green) reveals the presence of Nde1 in the nucleus of cells in the neocortical VZ. Notice the detection of Ndel1 in the cortical neurons of the Nde1$^{-/-}$ brain due to the cross reactivity of anti-NDE1/Nde1 to Ndel1. (**B**) Immunofluorescence confocal image of GFP-Nde1 (green) transfected HeLa cells showing nuclear targeting. Cell nuclei are highlighted by co-staining with Nucleoporin p62 at the nuclear envelope (red) as well as with Hoechst (blue). (**C**) Immunohistological analysis reveals enhanced nuclear Nde1 (red) in S phase neural progenitors (identified by Pcna foci in green). (**D**) Co-immunoprecipitation of the cohesin complex with Nde1. Myc-Nde1 was transfected in 293T cells and immunoprecipitated by the anti-myc 9E10 antibody or mouse IgG. The presence of SMC3, SMC1, and RAD21 in the Myc-Nde1 immunocomplex is shown by immunoblotting. (**E**) Binding of SMC3 NBD with Nde1. Flag-NBD of SMC3 was co-transfected with GFP-Nde1 in 293T cells and immunoprecipitated by the Flag antibody and mouse IgG. The presence of GFP-Nde1 and the absence SMC3 in the Flag-NBD immunocomplex are shown by immunoblotting. A diagram of the cohesin complex and the Nde1 binding domain (NBD) of SMC3 is included. (**F**) Flow cytometry analysis of cell cycle DNA content of 293T cells transfected with the vector

*Figure 7. Continued on next page*

Houlihan and Feng. eLife 2014;3:e03297. DOI: 10.7554/eLife.03297

*Figure 7. Continued*

control, Flag-NBD, and MycNde1-N1, a Nde1 N-terminal fragment that was previously shown to induce G2/M arrest by blocking Nde1 dimerization. (**G**) Nde1 co-complexes with cohesin and its associated chromatin remodeler SNF2h. Myc-Nde1 transfected in 293T was immunoprecipitated by the anti-myc 9E10 antibody using the mouse IgG as negative control. After the positive identification of core subunits of cohesin in the Myc-Nde1 immuno-complex, the immunoblots were re-probed with antibodies to several cohesin interacting proteins. SNF2h was consistently found to co-complex with Nde1 and cohesin.

The following figure supplements are available for figure 7:

**Figure supplement 1**. Blocking Nde1–cohesin interaction results in apoptosis.

**Figure supplement 2**. Increased genomic instability of Nde1 mutant MEFs.

## Discussion

We have elucidated a novel role of NDE1 in maintaining the brain genome. By interacting with cohesin and cohesin-associated chromatin remodeling complexes, Nde1 is specifically indispensible in mid-late S phase when DNA replication progresses through the heterochromatin. The loss of Nde1 function results in stalled DNA replication, DNA damage, and chromosomal instability, which together evoke DDRs, cell cycle exit, and p53-dependent apoptosis to eliminate neurons with genomic lesions (*Figure 8*). Nde1 mutations affect primary progenitors during their early fate restrictive differentiation to become neurons of cortical layer II/III.

### DNA replication and developmental cell differentiation

One of the most important observations presented in this study is that DNA damage and apoptosis in Nde1 mutant brain were spatially and temporally associated with the early neuronal differentiation of multipotent neural progenitors. The peak detection of γH2AX and CC3 in the developing cortex of Nde1 mutant mice was around E12.5. Levels of DNA damage and apoptosis were low, both during the early phase neural progenitor self renewal before E12 and when the bulk of cortical neurons were generated after E15. During the period when genomic lesions were detected at the peak level between E12 and E13, their abundance spatially correlated with the level of neurogenesis following the TNG. According to a recent genetic fate mapping study, at least a subset of progenitors become fate restricted to become upper layer cortical neurons before E13.5, even through these neurons are generated through terminal mitosis 3–4 days later (*Franco et al., 2012*). Therefore, the temporal-specific DNA damage that we observed in the Nde1 mutant brain matches closely with the time when primary neural progenitors become fate restricted to be layer II/III neurons. This phenotype is different from what have been seen in conditional brain mutation of genes known to play global roles in DNA metabolism and genome surveillance, such as Brca1, ATR, DNA ligase IV, XRCC4, TopBP1, and Cdh1 (*Frank et al., 2000*; *Gao et al., 2000*; *Pulvers and Huttner, 2009*; *Gatz et al., 2011*; *Lee et al., 2012*; *Eguren et al., 2013*). In these mutants, endogenous or gamma irradiation-induced DNA damage and p53-dependent apoptosis were shown to occur non-selectively in all progenitors that undergo active proliferation; abrogating p53 ameliorated the microcephaly but made little improvement to the structure of the mutant brain, which agree well with the wide-spread loss-of-function defects of these genes in many other organs. These previous studies demonstrated the importance of genome maintenance in brain development but did not reveal the brain specific endogenous origins of DNA damage and its underlying molecular mechanisms. In contrast, experimental findings of this study not only identified Nde1 as a key player for brain specific genome maintenance but also highlighted the principal source of DNA damage in brain development. Our data suggest that the strongest demand for genomic surveillance occurs in S phase and at heterochromatic DNA domains when neural progenitors undergo fate restrictive differentiation.

Heterochromatin constitutes a significant portion of the mammalian genome and is preferentially formed at chromosomal regions with high density of repetitive DNA elements, such as rRNA genes, centromeres, pericentromeres, and telomeres (*Lander et al., 2001*; *Grewal and Jia, 2007*; *Politz et al., 2013*). Compared to the actively transcribed euchromatin, heterochromatin is densely packed and traditionally considered transcriptionally inert. However, heterochromatin is known to produce RNA transcripts necessary for the establishment of heterochromatic states (*Bierhoff et al., 2014*), and these

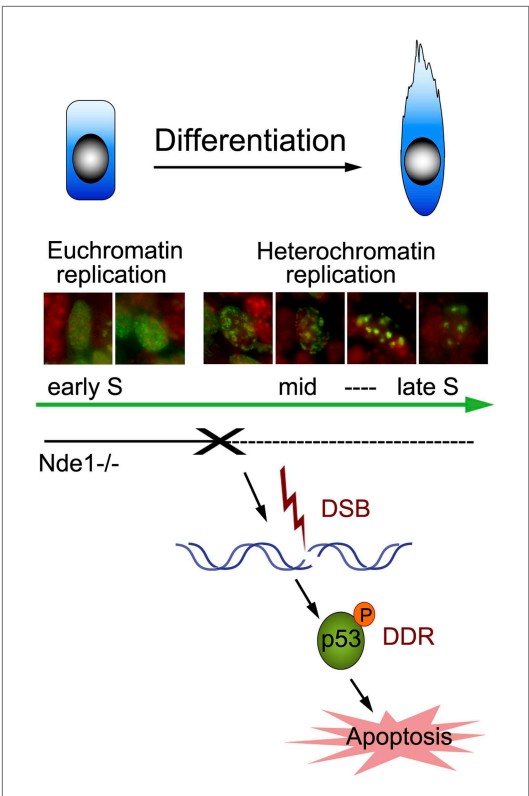

**Figure 8**. Nde1's role in maintaining genome integrity during early neural differentiation. During early neural differentiation, Nde1 mutant progenitors experience catastrophic DNA damage concurrent with mid-late S phase, when heterochromatic replication occurs. This evokes a DNA damage response which leads to the activation of p53-dependent apoptosis and results in the reduction of neurons in cortical layer II/III.

RNA transcripts were preferentially generated during the replication of the heterochromatin in late S phase (*Li et al., 2011*). Heterochromatin may not only influence gene expression profiles through regulating the higher-order chromatin structure, the repetitive sequences of heterochromatin are also hotspots for recombination which represent serious challenges to genomic integrity (*Peng and Karpen, 2008*; *Almouzni and Probst, 2011*). The proper maintenance of heterochromatin following the replication fork is indispensible for genomic stability by suppressing recombination and somatic mutations. The newly identified Nde1 partners cohesin and SNF2h are both known to participate in heterochromatin formation and function in multiple tissues. Therefore, further delineating the genomic target specifically co-regulated by Nde1and cohesin complex in neural progenitors will provide critical insight into how genomic quality is specifically controlled during neuronal differentiation.

In addition to the S phase, previous observation has shown that neocortical neurogenesis is governed by the length of G1 phase of the cell cycle (TG1) (*Lange et al., 2009*; *Pilaz et al., 2009*), which progressively increases over four-fold in mice from E11 to E17 (*Takahashi et al., 1995*; *Miyama et al., 1997*). While the lengthened G1 may permit increased gene expression and protein synthesis required for neurogenesis, another important task of G1 is to determine the timing of DNA replication and 'license' the replication origins through loading pre-replication complexes of DNA helicase at various chromosomal domains (*Dimitrova and Gilbert, 1999*; *Gilbert, 2010*; *Nordman and Orr-Weaver, 2012*; *Renard-Guillet et al., 2014*). In mammalian cells, over 30,000 origins are unutilized in each cell cycle; the spatiotemporal pattern of replication origin activation may depend on, as well as determine, a cell's differentiation state (*Gilbert, 2010*; *Nordman and Orr-Weaver, 2012*; *Rhind and Gilbert, 2013*). As the peak of S phase defect in Nde1 mutants at E12 to E13 overlaps with the developmental period, when Ts is at the top value and TG1 increased most drastically (*Takahashi et al., 1995*; *Miyama et al., 1997*), it is not unreasonable to speculate that the early phase of neuronal fate determination involves the reprogramming of chromatin state that initiates in G1 and executes in S phases. Thus, cell fate in cortical development may not merely be determined by turning on or off a handful of genes, but rather controlled at the genomic level by a nuclear global alteration of the chromatin state.

## Multifaceted function of Nde1 protein

The core Nde1 protein is approximately 40 kDa and comprises an N-terminal extended coiled-coil that shows remarkable evolutionary conservation. However, Nde1 is diverged structurally and functionally from its paralog Ndel1 in mammals outside of the coiled-coil and is primarily expressed in neural progenitors and hematopoietic cells (*Feng et al., 2000*). We have found that the native Nde1 protein in the developing mouse tissue is very insoluble and exists at a relatively low level, but genetic studies have shown that Nde1/NDE1 is an essential player for various pivotal aspects of the neural progenitor biology. It is likely that all native Nde1 molecules are fully occupied through binding to other cellular proteins. The multifaceted function of NDE1 as a molecular scaffold might be fulfilled by post-translational modifications or through undergoing reversible transitions between monomeric and polymeric forms

(*Feng and Walsh, 2004*; *Soares et al., 2012*), which facilitate its dynamic partition and reassembly into different molecular complexes. DNA replication is one of the most challenging processes that requires efficient and accurate molecular re-arrangements. During S phase, thousands of replication origins are utilized in a spatiotemporally defined pattern. At each licensed origin, helicase is loaded, replisome is assembled, nucleosomes are disassembled ahead of each replication fork, and resembled onto the two daughter genomes. Meanwhile, multiple histone chaperones, modification enzymes, and numerous chromatin remodelers are recruited to ensure the precise duplication of both the genome and the epigenome (*Alabert and Groth, 2012*; *Whitehouse and Smith, 2013*). Such a dynamic process requires a protein(s) like NDE1 to serve as a versatile platform to facilitate various protein–protein interactions. Thus, an evolutionarily advanced dynamic scaffold function during heterochromatin replication is consistent with a functional involvement of Nde1 in neural progenitor biology.

Nde1 has been implicated in the regulation of the cytoskeleton and dynein motors (*Lipka et al., 2013*). However, it is unclear how such regulation could underlie the tissue specific and spatiotemporally dependent phenotype of both human and mouse NDE1/Nde1 mutations nor the DNA replication defects described in this study. Our data suggest that instead Nde1 may be involved in assisting ATP-utilizing chromatin remodeling proteins such as SNF2h to move and reorganize nucleosomes along the DNA strand during heterochromatin replication. Replication forks in Nde1 mutant neural progenitors are paused or stalled at heterochromatic domains composed of centric/pericentric repeats. The integrity of centric/pericentric heterochromatic domains is essential not only for defining the chromatin state but also for the recruitment and the establishment of kinetochore complexes required for mitotic spindle formation and chromosome segregation in mitosis (*Ishii et al., 2008*; *DeLuca and Musacchio, 2012*; *Mankouri et al., 2013*). While a direct role of Nde1 in mitotic spindle organization remains plausible, we also believe that the skewed mitosis previously observed in Nde1$^{-/-}$ neural progenitors may, at least in part, originate from unresolved centromeric and pericentromeric structural defects arising during S phase. It remains to be further tested whether the mitotic spindle defect observed in Nde1 mutant progenitors is a primary cause for the defective cortical neurogenesis or a consequence of an ill-replicated genome.

## Genomic control of cerebral cortical evolution and function

The selective expression of Nde1 in neural progenitors and cells of hematopoietic lineage implies specific functional requirement of Nde1 in these cells. It is intriguing that the loss of NDE1 impairs the developing CNS much more profoundly than the hematopoietic system and other non-CNS tissues, though NDE1 mutations are associated with leukemogenesis (*Cavazzini et al., 2009*; *Van der Reijden et al., 2010*). In the developing CNS, the role of NDE1 is more essential in humans than in mice. Patients who lose both functional copies of *NDE1* show severe malformation of many brain structures, but the detectable brain anatomical defect of Nde1$^{-/-}$ mice is largely limited to the neocortex and specifically to layer II/III cortical neurons (*Feng and Walsh, 2004*; *Alkuraya et al., 2011*; *Bakircioglu et al., 2011*; *Guven et al., 2012*; *Paciorkowski et al., 2013*). While the mild phenotype in many key brain structures, such as the hippocampus, of Nde1 mice may be due to the functional redundancy of Nde1 with Ndel1, the common etiology of NDE1 mutation-induced genotoxicity in both neural and hematopoietic progenitors suggests that NDE1 may be important for establishing the evolutionarily increased cell diversity and function in both brain and blood. The cerebral cortex is the largest organ in terms of both cell number and functional diversity, and cortical layer II/III is expanded exponentially in evolution (*DeFelipe et al., 2002*; *Molnar et al., 2006*). The small and medium-size pyramidal neurons of cortical layer II/III are essential for functional connectivity between two cerebral hemispheres and among various cortical regions; they are essential for evolutionarily advanced higher order brain activities (*DeFelipe et al., 2002*; *Fame et al., 2011*; *Greig et al., 2013*). It is evident that even a slight perturbation to these neurons is likely to result in cognitive deficits. Therefore, it is conceivable that advanced novel molecular controls are gained in mammalian evolution to protect the genome of these neurons, although further study is required to elucidate the mechanism underlying such genome maintenance. Our data show that genomic damages during neurogenesis are primarily responsible for the cortical neuronal loss in Nde1 mutant brains. However, depending on the activity of DDRs, genomic damages may not necessarily result in altered brain structure as evidenced by the restored structure and layer II/III neurons in the Nde1$^{-/-}$Trp53$^{-/-}$ mutant. We believe that these data highlight the importance of genomic control of neuronal functions. Nde1 mutant cells presented an entire array of genotoxic features including DSBs and chromosomal lesions that would ordinarily lead to carcinogenesis. Nonetheless, even the Nde1$^{-/-}$Trp53$^{-/-}$ brain, which presumably carries the unresolved genomic

lesions to adulthood, was cancer-free. While the absence of brain tumor may be in line with our observation that genotoxicity in Nde1 mutant neural progenitors occurred when they become fate restricted along the path of differentiation towards post-mitotic neurons, it is still interesting to learn whether cortical neurons can truly forgive the genomic lesions they inherit from their precursors. With the advent of comprehensive genome-wide analysis, somatic mutations, especially copy number variants (CNVs) are found to be specifically abundant in human neurons and increasingly linked to neurodevelopmental diseases (*Grayton et al., 2012*; *McConnell et al., 2013*; *Moreno-De-Luca et al., 2013*). The *NDE1* gene is one of the 'hot spots' for CNVs at chromosome 16p13.11, of which deletions and duplications have been found to associate with a wide spectrum of developmental brain disorders including intellectual disability, epilepsy, autism, schizophrenia, and attention-deficit hyperactivity disorder (ADHD) (*Hebebrand et al., 1994*; *Gillberg, 1998*; *Sharp et al., 2006*; *Ullmann et al., 2007*; *Hannes et al., 2009*; *Heinzen et al., 2010*; *Mefford et al., 2010*). The genomic region span 16p13.11 encodes eight transcripts including MPV17L, C16orf45, KIAA0430, MYH11, C16orf63, ABCC1, and ABCC6 in addition to NDE1. Among these *NDE1* is the only gene that is known to be important for brain formation. Findings described in this study suggest that *NDE1* dosage alteration may result in secondary genomic lesions in cortical neurons. Therefore, even with the lack of neoplastic over-proliferation, brain developmental diseases may have a commonality with cancers as genomic mosaic genomic disorders. Although neurons with altered genome do not become cancerous due to their post-mitotic nature, they can be manifested by functional deficits of the brain in various forms. While further studies are required for delineating the causal relationship between *NDE1* gene dosage and compromised cortical functions, results from this study predict that genomic insults incurred during heterochromatic DNA replication during neural progenitor differentiation may underlie a large variety of developmental neurological and psychiatric disorders.

## Materials and methods

### Mouse strains

The Nde1 and Lis1 knockout mice have been described previously (*Feng and Walsh, 2004*; *Pawlisz et al., 2008*).The Trp53 knockout mice (Trp53tm1Tyj/J) were obtained from JaxMice (Bar Harbor, ME, stock # 002101). The Nde1–Trp53 double mutant mice were generated by standard genetic crosses. Mice used for this study were housed and bred according to the animal study protocol (protocol number 2012-1655) approved by IACUC committee of Northwestern University. All procedures were in compliance with the NIH Guide for Care and Use of Animals. For timed matings, the day of vaginal plug was considered E0.5.

### Immunohistology

Immunohistology studies were carried out as described (*Pawlisz et al., 2008*) on 12-µm frozen or 5-µm paraffin tissue sections. Neocortical coronal or transverse sections matched spatially at the mid-hemisphere level using the ganglionic eminence, the midline choroidal fissure, and the roof of the third ventricle as references were compared among littermates. The following antibodies were used: γH2AX, PH3, p53, NeuN (Millipore, Billerica, MA); phospho-p53 (Ser15), Rad50, γH2AX (Cell Signaling Tech, Beverly, MA); SMC3, Tbr2, BU1/75, Foxp2, NPM1 (Abcam, Cambridge, MA); PCNA, SMC1, 53PB1 (Novus Biologicals Littleton, CO); Cux1, DCX, BRCA1,Cdc25A (Santa Cruz, Dallas, TX); Pax6 (Thermo, Waltham, MA); B44 (BD Biosciences, San Jose, CA); phospho-vimentin 4A4 (MBL International, Woburn, MA); SNF2h (Active Motif, Carlsbad, CA); RAD21 (Bethyl Lab, Montgomery, TX); parvalbumin (Sigma, St. Louis, MO); Tuj1 (Covance, Princeton, NJ).

To detect Nde1 in mouse embryonic cortical progenitors, freshly dissected mouse embryos were embedded in OTC, frozen on dry ice, cryosectioned, fixed with 4% paraformaldehyde in PBS for 10 min, permeabilized with 0.1% Triton X-100, then immunostained with the NDE1 antibody (ProteinTech, Chicago, IL). As the NDE1 antibody cross-reacts with Ndel1 that is expressed highly in neurons but absent in neural progenitors, brain sections from Nde1$^{-/-}$ embryos were used as negative control so that Nde1-specific immunosignals in VZ progenitors could be distinguished. All experiments were repeated with at least three litters of mouse embryos; images from a representative experiment are shown.

### Nucleotide administration and cell cycle dynamic analysis

BrdU, IdU, or CldU (all from Sigma) were injected intraperitoneally (i.p.) to pregnant mice at 50 mg/kg to label embryos at E12.5. Labeled embryos were fixed, embedded, sectioned transversely,

and processed for immunohistology with antigen retrieval in citric acid-based antigen unmasking solution (Vector lab, Burlingame, CA). Double or triple immunofluorescence-stained coronal brain sections were imaged with a Leica CTR5000 fluorescence microscope equipped with a Qimage RETIGA 2000R digital camera under 20× or 40× objectives. Photographs from sections of the dorsal–medial cerebral wall at the mid-hemisphere level were taken. Fluorescent images were co-stained with Hoechst 33342 to identify cell nuclei. All images taken at different fluorescent channels were from the same focal-plane. At least three serial sections from three different litters for each genotype were analyzed.

## Quantification and statistical analysis

Quantitative analyses of immunosignals were performed with ImageJ or Adobe Photoshop CS4. Cell counts and comparisons between different genotypes were performed on mid-hemisphere neocortical sections. To avoid errors introduced by variable brain size, embryo shape, and embedding angles, sections were spatially matched using the ganglionic eminence and the midline choroidal fissure or the roof of the third ventricle, the anterior commissure, and the hippocampus as references for E12.5 or weaning age, respectively. The area of interest was specified in an approximately 400-micron length derived by measuring the distance along the ventricular surface in the dorsal and lateral pallium (between the medial pallium and the pallial–subpallial boundary) at E12.5 or in the neocortex at an approximate 45°-angle from the dorsal–ventral axis of mid-hemisphere level coronal sections at weaning age. All cells from the pial to ventricular surfaces were included in the analysis. Double positive cells were overlaid manually by color-coded dots in different layers. The number of cell counts was recorded using the measurement and analysis tools of either ImageJ or Photoshop CS4 and imported to Excel for quantitative statistical analysis and presentation. All results shown are mean ± SD from a minimum of three independent biological replicates. Statistical significance was estimated using the Student's $t$ test between two groups. Analysis of variance (ANOVA) was used for the comparison among groups with ≥3 categories (with SAS 9.4). When the overall F test from the ANOVA was significant ($p < 0.05$), Tukey-Kramer simulation-based adjusted p values were used for pairwise comparisons between the categories of the groups. Chi-Square test was used to compare proportions among categorical groups. Differences were considered significant with a $p < 0.05$.

## Plasmids, cell culture, transfection, and immunofluorescence

The plasmid encoding the Nde1 binding domain (NBD) of Smc3 was PCR amplified with primers ATG GATTAC AAGGATGACGACGATAAGAGACAGCAATCAGAA AAG and TTA GCGCTCCATACTTTTCTG, and a Flag tag was incorporated to the N-terminal end of the construct. The PCR product was cloned to pCR II TA cloning vector (Invitrogen), sequenced, and subcloned to pcDNA3.0 for mammalian cell expression. Cell culture, transfection, and immunofluorescence were performed as described (*Pawlisz and Feng, 2011*).

## Cortical lysates and immunoblotting

Cerebral cortices were dissected from mouse embryos and analyzed as described (*Pawlisz and Feng, 2011*). Experiments were repeated with samples from at least three litters; results from a representative experiment are shown.

## Comet assay

Comet assay for the assessment of DNA damage was performed according to the alkaline single-cell gel electrophoresis method as previously described (*Singh et al., 1988*) with minor modifications. Briefly, cortical progenitors were isolated in cold PBS with 20 mM EDTA from E12.5 mouse embryos, resuspended in 1% low melting point agar and PBS, and spread onto agar-coated slides. Cells on the slides were incubated in lysis buffer (2.5 M NaCl, 100 mM EDTA, 10 mM Tris base, 1% Triton X-100, and 10% DMSO) for 30 min at 4°C, followed by alkaline buffer (0.3 N NaOH and 1 mM EDTA) for 20 min at room temperature. After washing three times with 0.5× TBE, the cell-coated slides were electrophoresed at 0.7 volt/cm in 0.5× TBE, then neutralized, stained with DAPI, and analyzed under fluorescence microscope. Cells from Nde1 mutants and littermate controls were analyzed in parallel under the same experimental conditions, and comet tail length of over 300 randomly selected cells was scored. The assay was performed with more than three litters of Nde1, Lis1 double mutant embryos; results from representative experiments are shown.

## Immunoprecipitation

Immunoprecipitation was performed as described (*Pawlisz and Feng, 2011*). In a buffer with 25 mM HEPES, 150 mM NaCl, 10 mM NaF, 100 µM $Na_3VO_4$, 0.5% NP40, and 10% glycerol with the addition of protease inhibitors and 1 mM ATP.

## Flow cytometry

Flow cytometry analyses of cell cycle DNA content were performed as described (*Feng and Walsh, 2004*).

## Northern blotting analysis

Northern blotting analysis of Nde1 and Ndel1 expression was performed essentially as described (*Feng et al., 2000*). Briefly, 10 µg of total RNA extracted from various tissues of 3-month-old wild-type mice were loaded into each lane. Full-length coding cDNAs of mouse Nde1 and Ndel1 were used to probe the blot, respectively. Loading was normalized by the amount of 18S rRNA in each sample.

## Karyotyping

Mouse embryo fibroblasts were isolated and cultured as described (*Alkuraya et al., 2011*). To make the mitotic chromosome spreads, cells at passage 3, 5, and 10 grown on coverslips were arrested by 0.05 µg/ml colcemid for 5 hr. Karyotyping analyses were then performed as described (*Eves and Farber, 1981*). WT, Nde1$^{-/-}$ and Nde1$^{-/-}$Trp53$^{-/-}$ MEFs were cultured and analyzed in parallel. 80–120 mitotic spreads from MEFs derived from three different embryos were scored for each genotype. The seeding trypsinized embryo was counted as passage 0 and the first replating as passage 1.

## Acknowledgements

We would like to thank Alison Lanctot for discussions, suggestions, and reading of the manuscript. We thank Drs Sui Huang (Northwestern) and Supriya Prasanth (Univ. Illinois) for helpful comments and discussions. We thank Drs Yongchao Ma (Northwestern) and Noelle Dwyer (Univ. Virginia) for critical reading of the manuscript, and the RHLCC flow cytometry core facility for technical support. This work is supported by NICHD (R01HD56380) to YF.

## Additional information

### Funding

| Funder | Grant reference number | Author |
|---|---|---|
| National Institute of Child Health and Human Development | R01HD56380 | Yuanyi Feng |

The funder had no role in study design, data collection and interpretation, or the decision to submit the work for publication.

### Author contributions

SLH, Acquisition of data, Analysis and interpretation of data, Drafting or revising the article; YF, Conception and design, Acquisition of data, Analysis and interpretation of data, Drafting or revising the article

### Ethics

Animal experimentation: Mice used for this study were housed and bred according to the animal study protocol (protocol number 2012-1655) approved by IACUC committee of Northwestern University. All procedures were in compliance with NIH Guide for Care and Use of Animals.

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
