## [Decision Letter]

Thank you for sending your work entitled “Nde1 safeguards the brain genome during S phase of early neural progenitor differentiation” for consideration at *eLife.* Your article has been favorably evaluated by a Senior editor and 2 reviewers, one of whom is a member of our Board of Reviewing Editors.

The Reviewing editor and the other reviewers discussed their comments before we reached this decision, and the Reviewing editor has assembled the following comments to help you prepare a revised submission.

There are experimental and analytical details, interpretation of data and discussion points that will improve the manuscript:

1) At e12.5, there is a transverse neurogenetic gradient (Caviness et al TINS, 2009). Do the authors see this gradient in terms of onset of apoptosis in their material? The authors are imprecise in defining for the reader the specific neocortical region (anterior-posterior and medio-lateral) in which the analyses are done. Is this well-controlled from one animal to the next, across embryonic ages?

2) There is no information on sample sizes for most of the experiments. Additionally, the authors need to include a section in Methods regarding statistical analyses for the different experiments. There needs to be more information on counting methods used; if unbiased stereological methods were not used, then the authors need to justify their choice of method. For example, with regard to counting, what were the total number of layer II-III neurons and were they diminished both in relation to other cell types as well as to controls?

3) Was double labeling of nuclei with the different markers in brain sections was confirmed by z-stacking of the sections?

4) Based on Caviness et al, it appears that cell cycle 3-5 in the neurogenetic gradient in neocortex exhibits the greatest degree of DNA replication-associated damage. Why is this? Their conclusion is that this is the time (e11.5-13.5) of fate restriction for superficial cells. This corresponds to the TG1 middle phase, when G1 increases from ∼3 to ∼11 hr. Could this be the time of greatest vulnerability, as this is the time when protein synthesis and accumulation in G1 may govern cell fate decisions? The authors should incorporate a more thoughtful discussion of quantitative neuronogenesis concepts, which may allow them to build a testable model.

5) The hippocampus, a cortical structure, is normal in Nde1-/- mice. Any explanation for this?

6) The brains of Nde1 mutants appear to be globally smaller. If this is the case, and brain size limitations are not confined to the cortex, then the authors should discuss this as it relates to their hypotheses.

7) Nde1 is clearly not required for fate decisions – elimination of cell death via p53 is sufficient to restore normal cortical development. Really interesting, and should be emphasized. The authors point out in additional experiments that p53 elimination does not correct the genomic disruption caused by Nde1 deletion, yet prevents the disruption of production of layer 2-3 neurons. It's never really discussed well, the mechanism through which this might occur.

8) γH2AX double-labeling with 30 minute pulse-labeled progenitors should be quantified to get a sense of overlap, rather than relying just on staining (Figure 4). This is a valid criticism also for the B44/BU1/75 triple labeling in Figure 4. Authors note 'many cells were co-stained with γH2AX', but that is not accurate based on the images, in which there 6-8 co-labeled cells out of hundreds in the image.

9) The general model put forth implies that there is a greater level of aneuploidy in the adult cortex of double P53/Nde1 mutants. In situ aneuploidy staining methods are available and not difficult to employ. This data would be very helpful for improving the completeness of the manuscript.

10) Granted that Nde1's impact is fate and time-restricted, in contrast to other proteins (as noted in the Discussion), but the authors don't know the origins of this specificity. That is a shortfall of the study, but not fatal for impacting the importance of the study. However, to describe Nde1 as a 'guardian' of the brain genome is both overzealous and inaccurate. Such language needs to be toned down, to reflect or more limited understanding, based on the data presented here, of Nde1 in protecting against genome instability during a particular part of the cell cycle in a limited number of progenitor cells.

[Editors' note: further clarifications were requested prior to acceptance, as described below.]

Thank you for resubmitting your work entitled “The scaffold protein Nde1 safeguards the brain genome during S phase of early neural progenitor differentiation” for further consideration at *eLife*. Your revised article has been favorably evaluated by a Senior editor and a member of the Board of Reviewing Editors. The manuscript has been improved but there is one remaining issue that needs to be addressed before acceptance, as outlined below:

In the original decision letter, we noted that you did not use unbiased stereological counting; we asked you to justify not using this, but you did not. The counting method, given that you are expressing data as a ratio (percentage) of double-labeled cells, is reasonable. However, you do not specify the area dimensions of the regions counted. This is important, because the brain sizes are not the same. In addition, you did not use appropriate statistics. For example, the brain weight data has 8 genotype categories. You need to do an ANOVA to show a genotype affect before doing a post-hoc test to show which genotype. Individual t-test is not appropriate for your data. The same issue needs to be addressed for the double labeling studies in 3E and 3F. You have 3 genotypes, and thus, you need to report your analysis using ANOVA and then a post-hoc test for specific genotype effects.

---

## [Author Response]

1) At e12.5, there is a transverse neurogenetic gradient (Caviness et al TINS, 2009). Do the authors see this gradient in terms of onset of apoptosis in their material? The authors are imprecise in defining for the reader the specific neocortical region (anterior-posterior and medio-lateral) in which the analyses are done. Is this well-controlled from one animal to the next, across embryonic ages?

We have always observed the tight correlation of DNA damage and apoptosis with the spatiotemporal pattern in respect of the transverse neurogenetic gradient (TNG) in Nde1 mutant brains. Therefore, all immunohistological analysis in the study was performed with spatially matched brain sections among littermates. To formally address the question from the reviewer and present a more informative picture to readers, we serially sectioned and analyzed brains from 6 more Nde1-/- embryos. These new data which encompass the entire TNG are included in the in the revised Figure 1, Figure 1–figure supplement as well as their associated text. They should better demonstrate the correlation of DNA damage and apoptosis with the TNG between E12 and E13.

2) There is no information on sample sizes for most of the experiments. Additionally, the authors need to include a section in Methods regarding statistical analyses for the different experiments. There needs to be more information on counting methods used; if unbiased stereological methods were not used, then the authors need to justify their choice of method. For example, with regard to counting, what were the total number of layer II-III neurons and were they diminished both in relation to other cell types as well as to controls?

This study lasted more than 4 years during which hundreds of Nde1-/- and Nde1-/-Lis1+/-embryos were analyzed. All data from embryonic studies were obtained from at least 3 independent litters. The Nde1-p53 adult brain structure analysis was performed with 5 sets of age and genetic background matched mice. A more detailed description of these analyses and statistical methods has been added to the experimental methods section. We also included quantitative analysis of layer II/III neurons to the revised Figure 3 to show their specific reduction in Nde1-/- brain in relation to other neurons and their full restoration in Nde1-/-p53-/-mice.

3) Was double labeling of nuclei with the different markers in brain sections was confirmed by z-stacking of the sections?

All double /triple immunofluorescence stained tissue sections were co-stained with Hoechst 33342 to visualize nuclear DNA, thin sections (5 u or 12 u) were used, and images taken at different fluorescent channels were from the same focal-plane to assure that the correct cell was counted only once. We have noted this in the revised Methods section.

*4) Based on Caviness et al, it appears that cell cycle 3-5 in the neurogenetic gradient in neocortex exhibits the greatest degree of DNA replication-associated damage. Why is this? Their conclusion is that this is the time (e11.5-13.5) of fate restriction for superficial cells. This corresponds to the TG1 middle phase, when G1 increases from ∼3 to ∼11 hr. Could this be the time of greatest vulnerability, as this is the time when protein synthesis and accumulation in G1 may govern cell fate decisions? The authors should incorporate a more thoughtful discussion of quantitative neuronogenesis concepts, which may allow them to build a testable model*.

This excellent suggestion has been well received. As we indicated in the revised Discussion, we think that in addition to protein synthesis, increased TG1 may also relate to properly establishing the DNA replication pattern and chromatin state required for neural differentiation. This hypothesis explains our observation that the genome is more vulnerable during neuronal differentiation.

5) The hippocampus, a cortical structure, is normal in Nde1-/- mice. Any explanation for this?

As discussed in the revised manuscript, we think this is either due to the redundancy of Nde1 with Ndel1 or due to the evolutionarily advanced requirement or function of Nde1in genome maintenance since the neocortex, especially layer II/III, is evolutionarily novel, whereas the hippocampus is relatively less evolved from low to high vertebrates.

*6) The brains of Nde1 mutants appear to be globally smaller. If this is the case, and brain size limitations are not confined to the cortex, then the authors should discuss this as it relates to their hypotheses*.

The brain of Nde1homozygous mice is significantly smaller in the neocortex, but NDE1 homozygous mutation results in the severe reduction of many other brain structures in addition to the cortex. We believe this is in line with a more indispensable role of human NDE1 in more complex structures formed though evolution and we have addressed this in the revised Discussion.

*7) Nde1 is clearly not required for fate decisions* – *elimination of cell death via p53 is sufficient to restore normal cortical development. Really interesting, and should be emphasized. The authors point out in additional experiments that p53 elimination does not correct the genomic disruption caused by Nde1 deletion, yet prevents the disruption of production of layer 2-3 neurons. It's never really discussed well, the mechanism through which this might occur*.

Whether Nde1 is required for fate decision is dependent on the definition of “fate”. This study suggests that neuronal fate is not merely determined by the expression of a few markers, such as Cux1. Instead, we think the fate of cortical neurons should be further defined by many other fine characteristics including genomic signatures and chromatin states, although these fine characteristics need to be further elucidated in future studies. It is conceivable that restored Cux1+ neurons in the Nde1-/-p53-/- cortices may have functional deficits due to unrepaired genomic damage; they may have a different “fate” in terms of the genome or chromatin state. We believe further investigation of the function and refined cell biology features of these mutant neurons will shed light on a more advanced definition of cortical neuronal fate identities.

*8) γH2AX double-labeling with 30 minute pulse-labeled progenitors should be quantified to get a sense of overlap, rather than relying just on staining (*Figure 4*). This is a valid criticism also for the B44/BU1/75 triple labeling in*
Figure 4*. Authors note 'many cells were co-stained with γH2AX', but that is not accurate based on the images, in which there 6-8 co-labeled cells out of hundreds in the image*.

Quantifications of γH2AX in both BrdU transiently labeled (active S phase) and IdU+BrdU-(stalled S phase) have been added to the revised Figure 4 and the text of the revised manuscript. We hope this explains that γH2AX-BrdU transiently co-labeled cells indicate conceptually that DNA damage can occur in concurrence with DNA replication. However, DNA damage indicated by γH2AX is more frequently associated with stalled and defective S phase as indicated by Figure 4 and the revised text.

*9) The general model put forth implies that there is a greater level of aneuploidy in the adult cortex of double P53/Nde1 mutants. In situ aneuploidy staining methods are available and not difficult to employ. This data would be very helpful for improving the completeness of the manuscript*.

We are very thankful for this constructive suggestion. Showing genomic damage in Nde1-p53 double mutant brains directly has been one of our major efforts during this study. Several attempts have been made including a genome-wide study to identify aneuploidy and other chromosomal aberrations from adult cortical neurons using array-CGH. However, the presence of aneuploidy in wild type progenitors /neurons as well as the high heterogeneity of cortical cells and somatic mutations in Nde1 and Nde1-p53 brains prevented us from identifying consensus mutations. Although FISH can also be used to demonstrate aneuploidy, it requires sequence or chromosome specific DNA probes. Due to the random nature of somatic mutations and the putative involvement of heterochromatic repetitive sequences in the Nde1-p53 double mutant brain, it would be hard to pinpoint specific mutations and reach a statistically significant consensus conclusion with individual or a limited number of FISH probes. Therefore, we showed as clearly as was possible that both Nde1-/- and Nde1-/-;P53-/- progenitors experienced increased cell cycle and genomic stress through S phase, G2, and DDR analyses, and in particular, the increased degradation of Cdc25A (showed in Figure 6) is a hallmark of persistent genomic stress.

*10) Granted that Nde1's impact is fate and time-restricted, in contrast to other proteins (as noted in the Discussion), but the authors don't know the origins of this specificity. That is a shortfall of the study, but not fatal for impacting the importance of the study. However, to describe Nde1 as a 'guardian' of the brain genome is both overzealous and inaccurate. Such language needs to be toned down, to reflect or more limited understanding, based on the data presented here, of Nde1 in protecting against genome instability during a particular part of the cell cycle in a limited number of progenitor cells*.

We have eliminated “guardian” and other words which may not be accurately used in the text. However, it is clear that Nde1 deficiency results in genomic damage in the brain. The tissue, cell type and developmental specificity of the phenotype suggest a mechanism specific to evolutionarily advanced complex tissue with high cell diversity. We believe the study provides a new entry point, rather than conclusion, for further investigating the neuronal fate control at the genomic level.

[Editors' note: further clarifications were requested prior to acceptance, as described below.]

*In the original decision letter, we noted that you did not use unbiased stereological counting; we asked you to justify not using this, but you did not. The counting method, given that you are expressing data as a ratio (percentage) of double-labeled cells, is reasonable. However, you do not specify the area dimensions of the regions counted. This is important, because the brain sizes are not the same. In addition, you did not use appropriate statistics. For example, the brain weight data has 8 genotype categories. You need to do an ANOVA to show a genotype affect before doing a post-hoc test to show which genotype. Individual t-test is not appropriate for your data. The same issue needs to be addressed for the double labeling studies in 3E and 3F. You have 3 genotypes, and thus, you need to report your analysis using ANOVA and then a post-hoc test for specific genotype effects*.

Thank you very much for your continued effort to make this a better paper. I am sorry that we did not explain the reason for not using unbiased stereological counting. The cerebral cortex is a well-organized and laminated structure in both embryonic and adult brains, yet its cellular organization is spatially dependent and varies in different brain regions. We felt that more accurate counts for both embryonic ventricular zone progenitors and postnatal cortical neurons may be obtained from spatially well-matched brain sections than unbiased random cell samplings, and we tried to match the tissue sections among different genotypes or individual brains using structures not altered by Nde1 mutation as landmarks. These details, including areas selected for cell counting and their dimension, are now added to the experimental method.

We very much appreciate your correction on our method of statistical analysis. We have reassessed statistical significance of all data from multiple genotype categories using ANOVA followed by post-hoc pairwise comparisons. The revised statistical analysis and results are included in the Methods and figure legends.